# WHY ADVERSARIAL TRAINING CAN HURT ROBUST ACCURACY

**Jacob Clarysse[1], Julia Hörrmann[2], Fanny Yang[1]**
1. Department of Computer Science, ETH Zürich
2. Department of Mathematics, ETH Zürich
{jacob.clarysse;fan.yang}@inf.ethz.ch;
{julia.hoerrmann}@stat.math.ethz.ch

## ABSTRACT

Machine learning classifiers with high test accuracy often perform poorly under adversarial perturbations. It is commonly believed that adversarial training alleviates this issue. In this paper, we demonstrate that, surprisingly, the opposite can be true for a natural class of perceptible perturbations — even though adversarial training helps when enough data is available, it may in fact hurt robust generalization in the small sample size regime. We first prove this phenomenon for a high-dimensional linear classification setting with noiseless observations. Using intuitive insights from the proof, we could find perturbations on standard image datasets for which this behavior persists. Specifically, it occurs for perceptible perturbations that effectively reduce class information such as object occlusions or corruptions.

## 1 INTRODUCTION

Today's best-performing classifiers are vulnerable to adversarial attacks Goodfellow et al. (2015); Szegedy et al. (2014) and exhibit high *robust error*: for many inputs, their predictions change under adversarial perturbations, even though the true class stays the same. Such content-preserving (Gilmer et al., 2018), consistent (Raghunathan et al., 2020) attacks can be either perceptible or imperceptible. For image datasets, most work to date studies imperceptible attacks that are based on perturbations with limited strength or *attack budget*. These include bounded $\ell_p$-norm perturbations (Goodfellow et al., 2015; Madry et al., 2018; Moosavi-Dezfooli et al., 2016), small transformations using image processing techniques (Ghiasi et al., 2019; Zhao et al., 2020; Laidlaw et al., 2021; Luo et al., 2018) or nearby samples on the data manifold (Lin et al., 2020; Zhou et al., 2020). Even though they do not visibly change the image by definition, imperceptible attacks can often successfully fool a learned classifier.

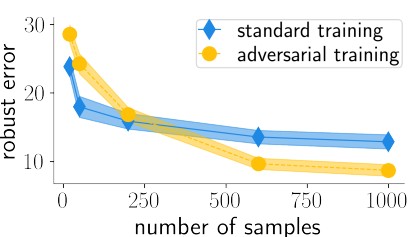

Figure 1: On the Waterbirds dataset attacked by the adversarial illumination attack, adversarial training (yellow) yields higher robust error than standard training (blue) when the sample size is small, even though it helps for large sample sizes and in a setting where the standard error of standard training is small. (see App. D for details).

On the other hand, perturbations that naturally occur and are physically realizable are commonly perceptible. Some perceptible perturbations specifically target the object to be recognized: these include occlusions (e.g. stickers placed on traffic signs (Eykholt et al., 2018) or masks of different sizes that cover important features of human faces (Wu et al., 2020)) or corruptions that are caused by the image capturing process (animals that move faster than the shutter speed or objects that are not well-lit, see Figure 2). Others transform the whole image and are not confined to the object itself, such as rotations, translations or corruptions Engstrom et al. (2019); Kang et al. (2019). In this paper, we refer to such perceptible attacks as *directed attacks*. In contrast to other attacks, they effectively reduce useful class information in the input for any model, without necessarily changing the true label — we say that they are *directed* and *consistent*, more formally defined in Section 2. For example, a stop sign with a small sticker could partially cover the text without losing its semantic meaning. Similarly, a flying bird captured with a long exposure time can induce motion blur in the final image without becoming unrecognizable to the observer.

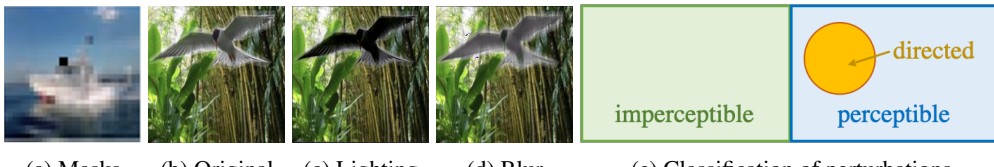

|  (a) Masks | (b) Original | (c) Lighting | (d) Blur | (e) Classification of perturbations |

Figure 2: Examples of directed attacks on CIFAR10 and the Waterbirds dataset. In Figure 2a, we corrupt the image with a black mask of size $2 \times 2$ and in Figure 2c and 2d we change the lighting conditions (darkening) and apply motion blur on the bird in the image respectively. All perturbations reduce the information about the class in the images: they are the result of directed attacks. (e) Directed attacks are a subset of perceptible attacks.

In the literature so far, it is widely acknowledged that adversarial training with the same perturbation type and budget as during test time often achieves significantly lower robust error than standard training (Madry et al., 2018; Zhang et al., 2019; Bai et al., 2021).

In contrast, we show that adversarial training not only increases standard error (Zhang et al., 2019; Tsipras et al., 2019; Stutz et al., 2019; Raghunathan et al., 2020), but surprisingly, in the low sample regime,

*adversarial training may even increase the robust error compared to standard training!*

Figure 1 illustrates the main message of our paper on the Waterbirds dataset: Although adversarial training with directed attacks outperforms standard training when enough training samples are available, it is inferior when the sample size is small (but still large enough to obtain a small standard test error).

Our contributions are as follows:

- We prove that, almost surely, adversarially training a linear classifier on separable data yields a monotonically increasing robust error as the perturbation budget grows. We further establish high-probability non-asymptotic lower bounds on the robust error gap between adversarial and standard training.
- Our proof provides intuition for why this lower bound on the gap is particularly large for directed attacks in the low sample regime.
- We observe empirically for different directed attacks on real-world image datasets that this behavior persists: adversarial training for directed attacks hurts robust accuracy when the sample size is small.

## 2 ROBUST CLASSIFICATION

We first introduce our robust classification setting more formally by defining the notions of adversarial robustness, directed attacks and adversarial training used throughout the paper.

**Adversarially robust classifiers**  For inputs $x \in \mathbb{R}^d$, we consider multi-class classifiers associated with parameterized functions $f_\theta : \mathbb{R}^d \to \mathbb{R}^K$ if $K > 2$ and $f_\theta : \mathbb{R}^d \to \mathbb{R}$ if $K = 2$, where $K$ is the number of labels. For example, $f_\theta(x)$ could be a linear model (as in Section 3) or a neural network (as in Section 4). The output label predictions are obtained by $h(f_\theta(x)) = \text{sign}(f_\theta(x))$ for $K = 2$ and $h(f_\theta(x)) = \arg\max_{k \in \{1,..,K\}} f_\theta(x)_k$ for $K > 2$.

In order to convince practitioners to use machine learning models in the wild, it is key to demonstrate that they exhibit robustness. One kind of robustness is that they do not change prediction when the input is subject to consistent perturbations, which are small class-preserving perturbations. Mathematically speaking, for the underlying joint data distribution $\mathbb{P}$, the model should have a small $\epsilon_{te}$-*robust error*, defined as

$$\text{Err}(\theta; \epsilon_{\text{te}}) := \mathbb{E}_{(x,y)\sim\mathbb{P}} \max_{x' \in T(x;\epsilon_{\text{te}})} \ell(f_\theta(x'), y), \tag{1}$$

where $\ell$ is 0 if the class determined by $h(f_\theta(x))$ is equal to $y$ and 1 otherwise. Further, $T(x; \epsilon_{te})$ indicates a perturbation set around $x$ of a certain transformation type with size $\epsilon_{test}$. Note that

the *(standard) error* $\mathbb{E}_{(x,y)\sim\mathbb{P}}\ell(f_\theta(x),y)$ of a classifier corresponds to $\text{Err}(\theta;0)$ – the robust error evaluated at $\epsilon_{\text{te}} = 0$.

**Directed attacks**    The inner maximization in Equation 1 is often called the adversarial *attack* of the input $x$ for the model $f_\theta$ and the corresponding solution is referred to as the *adversarial example*. In this paper, we consider *directed attacks* that effectively reduce the information about the true classes, with image-based examples depicted in Figure 2. For linear classification, we analyze directed attacks in the form of additive perturbations that are constrained to the direction of the optimal decision boundary (see details in Section 3.1). In particular, note that the set of directed perturbations is restricted to directions attacking the Bayes optimal classifier.

**Adversarial training**    A common approach to obtain classifiers with a good robust accuracy is to minimize the training objective $\mathcal{L}_{\epsilon_{\text{tr}}}$ with a surrogate robust classification loss $L$

$$\mathcal{L}_{\epsilon_{\text{tr}}}(\theta) := \frac{1}{n}\sum_{i=1}^{n} \max_{x_i'\in T(x_i;\epsilon_{\text{tr}})} L(f_\theta(x_i')y_i), \tag{2}$$

also called *adversarial training*. In practice, we often use the cross entropy loss $L(z) = \log(1 + e^{-z})$ and minimize the robust objective by using first order optimization methods such as (stochastic) gradient descent. SGD is also the algorithm that we focus on in both the theoretical and experimental sections. When the desired type of robustness is known in advance, it is standard practice to use the same perturbation set for training as for testing, i.e. $T(x;\epsilon_{\text{tr}}) = T(x;\epsilon_{\text{te}})$. For example, Madry et al. (2018) show that the robust error sharply increases for $\epsilon_{\text{tr}} < \epsilon_{\text{te}}$. In this paper, we demonstrate that for directed attacks in the small sample size regime, in fact, the opposite is true.

## 3 THEORETICAL RESULTS

In this section, we prove for linear functions $f_\theta(x) = \theta^\top x$ that in the case of directed attacks, robust generalization deteriorates with increasing $\epsilon_{\text{tr}}$. The proof, albeit in a simple setting, provides explanations for why adversarial training fails in the high-dimensional regime for such attacks.

### 3.1 SETTING

We now introduce the precise linear setting used in our theoretical results.

**Data model**    We assume that the ground truth and hypothesis class are given by linear functions $f_\theta(x) = \theta^\top x$ and the sample size $n$ is lower than the ambient dimension $d$ minus one. The generative distribution $\mathbb{P}_r$ is similar to (Tsipras et al., 2019; Nagarajan & Kolter, 2019): The label $y \in \{+1, -1\}$ is drawn with equal probability and the covariate vector is sampled as $x = [y\frac{r}{2}, \tilde{x}]$ with the random vector $\tilde{x} \in \mathbb{R}^{d-1}$ drawn from a standard normal distribution, i.e. $\tilde{x} \sim \mathcal{N}(0, \sigma^2 I_{d-1})$. We would like to learn a classifier that has low robust error by using a dataset $D = (x_i, y_i)_{i=1}^{n}$ with $n$ i.i.d. samples from $\mathbb{P}_r$. Intuitively, the separation distance $r$ reflects the signal strength of the data distribution.

Notice that the distribution $\mathbb{P}_r$ is noiseless: for a given input $x$, the label $y = \text{sign}(x_{[1]})$ is deterministic. Further, the Bayes optimal linear classifier (also referred to as the *ground truth*) is parameterized by the first standard coordinate vector, $\theta^\star = e_1$.[1] By definition, the ground truth is robust against all perturbations that do not change the sign in the first coordinate of the sample, i.e. consistent perturbations, and hence so is the optimal robust classifier.

**Directed attacks**    In this paper, we focus on consistent directed attacks that by definition efficiently concentrate their attack budget to reduce the class information. For our linear setting this information lies in the first entry. Hence, we can model such attacks by additive perturbations in the first dimension

$$T(x;\epsilon) = \{x' = x + \delta \mid \delta = \beta e_1 \text{ and } -\epsilon \leq \beta \leq \epsilon\}. \tag{3}$$

Note that this attack is always in the direction of the signal dimension, i.e. the Bayes optimal classifier or equivalently the ground truth. Furthermore, when $\epsilon < \frac{r}{2}$, it is a consistent directed attack. Observe how this is different from $\ell_p$-attacks — an $\ell_p$ attack, depending on the model, may add a perturbation that only has a very small component in the signal direction.

---

[1]Note that the result more generally holds for non-sparse models that are not axis aligned by way of a simple rotation $z = Ux$. In that case the distribution is characterized by $\theta^\star = u_1$, where $u_1$ is the first column vector of $U$, and a rotated Gaussian in the $d - 1$ dimensions orthogonal to $\theta^\star$.

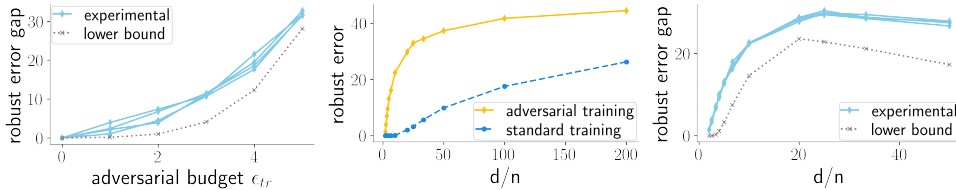

(a) Robust error increase with $\epsilon_{tr}$  (b) Standard-adversarial training  (c) Effect of overparameterization

Figure 3: Experimental verification of Theorem 3.1. (a) We set $d = 1000$, $r = 12$ and $n = 50$. The robust error gap between standard and adversarial training as a function of the adversarial budget $\epsilon_{tr} = 5$ independent experiments (blue) and the lower bound given in Theorem 3.1 (gray). In (b) and (c), we set $d = 10000$ and vary the number of samples $n$. (b) The robust error of standard and adversarial training with $\epsilon_{tr} = 4.5$. (c) The error gap and the lower bound of Theorem 3.1. For more experimental details see Appendix C.

**Robust max-$\ell_2$-margin classifier**  We study a classifier that is the solution of running gradient descent on the adversarial logistic loss. A long line of work (Soudry et al., 2018; Ji & Telgarsky, 2019; Chizat & Bach, 2020; Nacson et al., 2019; Liu et al., 2020) studies the implicit bias of (S)GD on the (standard) logistic loss and separable data. In particular, they show directional convergence to the max-margin solution. For the adversarial logistic loss and linear models in particular, (S)GD converges to the robust max-$\ell_2$-margin solution (Li et al., 2020),

$$\widehat{\theta}^{\epsilon_{tr}} := \arg\max_{\|\theta\|_2 \leq 1} \min_{i \in [n], x'_i \in T(x_i; \epsilon_{tr})} y_i \theta^\top x'_i. \tag{4}$$

Even though our result is proven for the max-$\ell_2$-margin classifier, it can easily be extended to other interpolators.

## 3.2  MAIN RESULTS

We are now ready to characterize the $\epsilon_{te}$-robust error as a function of $\epsilon_{tr}$, the separation $r$, the dimension $d$ and sample size $n$ of the data. In the theorem statement we use the following quantities

$$\varphi_{\min} = \frac{\sigma}{r/2 - \epsilon_{te}} \left( \sqrt{\frac{d-1}{n}} - \left(1 + \sqrt{\frac{2\log(2/\delta)}{n}}\right) \right)$$

$$\varphi_{\max} = \frac{\sigma}{r/2 - \epsilon_{te}} \left( \sqrt{\frac{d-1}{n}} + \left(1 + \sqrt{\frac{2\log(2/\delta)}{n}}\right) \right)$$

that arise from concentration bounds for the singular values of the random data matrix. Further, let $\tilde{\epsilon} := \frac{r}{2} - \frac{\varphi_{\max}}{\sqrt{2}}$ and denote by $\Phi$ the cumulative distribution function of a standard normal.

**Theorem 3.1.** *Assume $d - 1 > n$. For test samples from $\mathbb{P}_r$, perturbation set type $T$ as in Equation 3 and any $0 \leq \epsilon_{te} < \frac{r}{2}$, the following holds for the $\epsilon_{te}$-robust error of the classifier (Equation 1) resulting from $\epsilon_{tr}$-adversarial training:*

1. *The $\epsilon_{te}$-robust error of the $\epsilon_{tr}$-robust max-margin estimator reads*

$$Err(\widehat{\theta}^{\epsilon_{tr}}; \epsilon_{te}) = \Phi\left( -\frac{\left(\frac{r}{2} - \epsilon_{tr}\right)}{\tilde{\varphi}} \right) \tag{5}$$

   *for a random quantity $\tilde{\varphi} > 0$ depending on $\sigma, r, \epsilon_{te}$ and is hence strictly increasing in the adversarial training budget $\epsilon_{tr}$.*

2. *With probability at least $1 - \delta$, we further have $\varphi_{min} \leq \tilde{\varphi} \leq \varphi_{max}$ and the following lower bound on the robust error increase by adversarially training with size $\epsilon_{tr}$*

$$Err(\widehat{\theta}^{\epsilon_{tr}}; \epsilon_{te}) - Err(\widehat{\theta}^0; \epsilon_{te}) \geq \Phi\left( \frac{r/2}{\varphi_{min}} \right) - \Phi\left( \frac{r/2 - \min\{\epsilon_{tr}, \tilde{\epsilon}\}}{\varphi_{min}} \right). \tag{6}$$

The proof can be found in Appendix A and primarily relies on estimation of singular values of high-dimensional matrices. Note that the theorem holds for any $0 \leq \epsilon_{te} < \frac{r}{2}$ and hence also directly

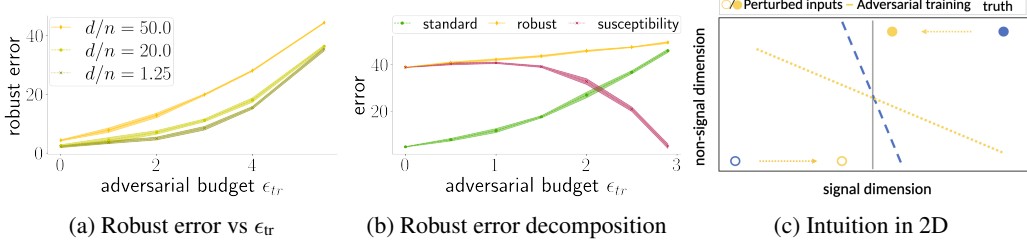

(a) Robust error vs $\epsilon_{\text{tr}}$      (b) Robust error decomposition      (c) Intuition in 2D

Figure 4: (a) We set $d = 1000$ and $r = 12$. The robust error as a function of the adversarial training budget $\epsilon_{\text{tr}}$ for different $d/n$. (b) The robust error decomposition into susceptibility and standard error as a function of the adversarial budget $\epsilon_{\text{tr}}$. Full experimental details can be found in Section C. (c) 2D illustration providing intuition for the linear setting. The effect of adversarial training with directed attacks is captured in the yellow dotted lines: adversarially perturbed training points move closer to the true boundary which in turn tilts the decision boundary more heavily in the wrong direction.

applies to the standard error by setting $\epsilon_{\text{te}} = 0$. In Figure 3, we empirically confirm the statements of Theorem 3.1 by performing multiple experiments on synthetic datasets as described in Subsection 3.1 with different choices of $d/n$ and $\epsilon_{\text{tr}}$. In the first statement, we prove that for small sample-size ($n < d - 1$) noiseless data, almost surely, the robust error increases monotonically with adversarial training budget $\epsilon_{\text{tr}} > 0$. In Figure 3a, we plot the robust error gap between standard and adversarial logistic regression as a function of the adversarial training budget $\epsilon_{\text{tr}}$ for 5 runs.

The second statement establishes a simplified lower bound on the robust error increase for adversarial training (for a fixed $\epsilon_{\text{tr}} = \epsilon_{\text{te}}$) compared to standard training. In Figures 3a and 3c, we show how the lower bound closely predicts the robust error gap in our synthetic experiments. Furthermore, by the dependence of $\varphi_{\min}$ on the overparameterization ratio $d/n$, the lower bound on the robust error gap is amplified for large $d/n$. Indeed, Figure 3c shows how the error gap increases with $d/n$ both theoretically and experimentally. However, when $d/n$ increases above a certain threshold, the gap decreases again, as standard training fails to learn the signal and yields a high error (see Figure 3b).

### 3.3 Proof intuition

The reason that adversarial training hurts robust generalization is based on an extreme robust vs. standard error trade-off. We now provide intuition for the effect of directed attacks and the low sample regime on the $\epsilon_{\text{tr}}$-robust max-$\ell_2$-margin solution by decomposing the robust error $\text{Err}(\theta; \epsilon_{\text{te}})$. Notice that $\epsilon_{\text{te}}$-robust error $\text{Err}(\theta; \epsilon_{\text{te}})$ can be written as the probability of the union of two events: the event that the classifier based on $\theta$ is wrong and the event that the classifier is susceptible to attacks:

$$\text{Err}(\theta; \epsilon_{\text{te}}) = \mathbb{E}_{x,y \sim \mathbb{P}}\left[\mathbb{I}\{yf_\theta(x) < 0\} \vee \max_{x' \in T(x; \epsilon_{\text{te}})} \mathbb{I}\{f_\theta(x)f_\theta(x') < 0\}\right] \leq \text{Err}(\theta; 0) + \text{Susc}(\theta; \epsilon_{\text{te}})$$

(7)

where $\text{Susc}(\theta; \epsilon_{\text{te}})$ is the expectation of the maximization term in Equation 7. $\text{Susc}(\theta; \epsilon_{\text{te}})$ represents the $\epsilon_{\text{te}}$-*attack-susceptibility* of a classifier induced by $\theta$ and $\text{Err}(\theta; 0)$ its standard error. In our linear setting, we can lower bound Equation 7 by $\text{Err}(\theta; 0) + \frac{1}{2}\text{Susc}(\theta; \epsilon_{\text{te}})$. Hence, Equation 7 suggests that the robust error can only be small if both the standard error and susceptibility are small. In Figure 4b, we plot the decomposition of the robust error in standard error and susceptibility for adversarial logistic regression with increasing $\epsilon_{\text{tr}}$. We observe that increasing $\epsilon_{\text{tr}}$ increases the standard error too drastically compared to the decrease in susceptibility, leading to a drop in robust accuracy. For completeness, in Appendix B, we provide upper and lower bounds for the susceptibility score. We now explain why, in the small-sample size regime, adversarial training with directed attacks 3 may increase standard error to the extent that it dominates the decrease in susceptibility.

A key observation is that the robust max-$\ell_2$-margin solution of a dataset $D = \{(x_i, y_i)\}_{i=1}^n$ maximizes the minimum margin that reads $\min_{i \in [n]} y_i \theta^\top(x_i - y_i\epsilon_{\text{tr}}|\theta_{[1]}|e_1)$, where $\theta_{[i]}$ refers to the $i$-th entry of vector $\theta$. Therefore, it simply corresponds to the max-$\ell_2$-margin solution of the dataset shifted towards the decision boundary $D_{\epsilon_{\text{tr}}} = \{(x_i - y_i\epsilon_{\text{tr}}|\widehat{\theta}_{[1]}^{\epsilon_{\text{tr}}}|e_1, y_i)\}_{i=1}^n$. Using this fact, we obtain a closed-form expression of the (normalized) max-margin solution 4 as a function of $\epsilon_{\text{tr}}$ that reads

$$\widehat{\theta}^{\epsilon_{\text{tr}}} = \frac{1}{(r - 2\epsilon_{\text{tr}})^2 + 4\tilde{\gamma}^2}\left[r - 2\epsilon_{\text{tr}}, 2\tilde{\gamma}\tilde{\theta}\right],$$

(8)

where $\|\tilde{\theta}\|_2 = 1$ and $\tilde{\gamma} > 0$ is a random quantity associated with the max-$\ell_2$-margin solution of the $d - 1$ dimensional Gaussian inputs orthogonal to the signal direction (see Lemma A.1 in Section A).

In high dimensions, with high probability any two Gaussian random vectors are far apart – in our distributional setting, this corresponds to the vectors being far apart in the non-signal directions. In Figure 4c, we illustrate the phenomenon using a 2D cartoon, where the few samples in the dataset are all far apart in the non-signal direction. We see how shifting the dataset closer to the true decision boundary, may result in a max-margin solution (yellow) that aligns much worse with the ground truth (gray), compared to the estimator learned from the original points (blue). Even though the new (robust max-margin) classifier (yellow) is less susceptible to attacks in the signal dimension, it also uses the signal dimension less. Mathematically, this is reflected in the expression of the max-margin solution in Equation 8: We see that the first (signal) dimension is used less as $\epsilon_{\text{tr}}$ increases.

## 3.4 GENERALITY OF THE RESULTS

In this section we discuss how Theorem 3.1 might generalize to other perturbation sets and models.

**Signal direction is known** The type of additive perturbations used in Theorem 3.1, defined in Equation 3, is explicitly constrained to the direction of the true signal. This choice is reminiscent of corruptions where every possible perturbation in the set is directly targeted at the object to be recognized, such as motion blur of moving objects. Such corruptions are also studied in the context of domain generalization and adaptation (Schneider et al., 2020).

Directed attacks in general, however, may also consist of perturbation sets that are only strongly biased towards the true signal direction. They may find the true signal direction only when the inner maximization is exact. The following corollary extends Theorem 3.1 to small $\ell_1$-perturbations

$$T(x; \epsilon) = \{x' = x + \delta \mid \|\delta\|_1 \le \epsilon\}, \tag{9}$$

for $0 < \epsilon < \frac{r}{2}$ that reflect such attacks. We state the corollary here and give the proof in Appendix A.

**Corollary 3.2.** *Theorem 3.1 also holds for 4 with perturbation sets defined in 9.*

The proof uses the fact that the inner maximization effectively results in a sparse perturbation equivalent to the attack resulting from the perturbation set defined in Equation 3.

**Other models** Motivated by the implicit bias results of (stochastic) gradient descent on the logistic loss, Theorem 3.1 is proven for the max-$\ell_2$-margin solution. We would like to conjecture that for the data distribution in Section 3, adversarial training can hurt robust generalization also for other models with zero training error (*interpolators* in short). For example, Adaboost is a widely used algorithm that converges to the max-$\ell_1$-margin classifier (Telgarsky, 2013). One might argue that for a sparse ground truth, the max-$\ell_1$-margin classifier should (at least in the noiseless case) have the right inductive bias to alleviate large bias in high dimensions. Hence, in many cases the (sparse) max-$\ell_1$-margin solution might align with the ground truth for a given dataset. However, we conjecture that even in this case, the *robust* max-$\ell_1$-margin solution would be misled to choose a wrong sparse solution. This can be seen with the help of the cartoon illustration in Figure 4c.

## 4 REAL-WORLD EXPERIMENTS

In this section, we demonstrate that the proof intuition of the linear case may generalize to more complex models. Specifically, the insights from Section 3 helped us to identify realistic directed attacks on standard image datasets for which adversarial training hurts robust accuracy in the low sample regime. In what follows, we present experimental results for corruption attacks on the Waterbirds dataset. Due to space constraints, results on the mask attacks on CIFAR-10 can be found in Appendix E. The corresponding experimental details and more results on other additional image datasets (such as the hand gestures dataset) can be found in Appendices D, E and F.

## 4.1 DATASETS AND MODELS

We consider three datasets: the Waterbirds dataset, CIFAR-10 and a hand gesture datasets. Due to space constraints, we describe CIFAR-10 and the hand gesture dataset in Appendix E and F. Apart from CIFAR-10 and the hand gesture dataset, we build a new version of the Waterbirds dataset, consisting of images of water- and landbirds of size $256 \times 256$ and labels that distinguish the two

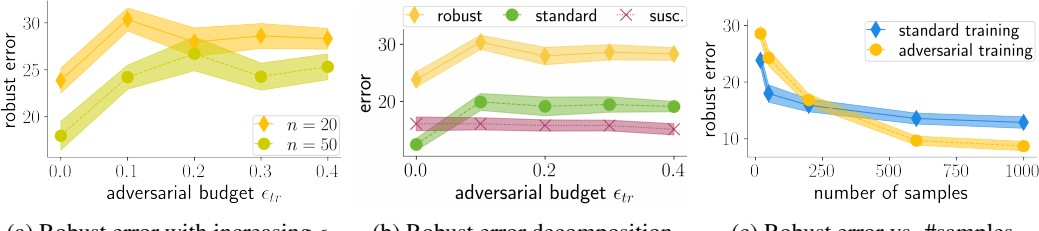

(a) Robust error with increasing $\epsilon_{\text{tr}}$    (b) Robust error decomposition    (c) Robust error vs. #samples

Figure 5: Experiments on the Waterbirds dataset considering the adversarial illumination attack with $\epsilon_{\text{te}} = 0.3$. We plot the mean and standard deviation of the mean of several independent experiments. (a) The robust error increases with larger $\epsilon_{\text{tr}}$ in the low sample size regime. (b) We set $n = 20$ and plot the robust error decomposition as in Equation 7 with increasing $\epsilon_{\text{tr}}$. While the susceptibility decreases slightly, the increase in standard error is much more severe, resulting in an increase in robust error. (c) The robust error of standard training and adversarial training as a function of the number of samples, where the smallest sample size still yields small ($< 10\%$) standard test error for standard training. While adversarial training hurts for small sample sizes, it helps for larger sample sizes. For more experimental details see App. D.

types of birds. Using code provided by Sagawa et al. (2020), we construct the dataset as follows: First, we sample equally many water- and landbirds from the CUB-200 dataset (Welinder et al., 2010). Then, we segment the birds and paste them onto a background image that is randomly sampled (without replacement) from the Places-256 dataset (Zhou et al., 2017). Also, following the choice of Sagawa et al. (2020), we use as models a ResNet50 and a ResNet18 that were both pretrained on ImageNet and achieve near perfect standard accuracy. In Appendix D, we complement the results of this section by reporting the results of similar experiments with different architectures.

## 4.2 IMPLEMENTATION OF THE DIRECTED ATTACKS

In this section, we consider two attacks on the Waterbirds dataset: motion blur and adversarial illumination as depicted in Figure 2. In Appendix E, we also discuss the mask attack, which should mimic occlusions of objects in images that are physically realizable (Eykholt et al., 2018; Wu et al., 2020). On the other hand, motion blur may arise naturally when photographing fast moving objects with a slow shutter speed. Lastly, adversarial illumination may result from adversarial lighting conditions. Next, we describe the motion blur and adversarial illumination attacks in more detail.

**Motion blur** For the Waterbirds dataset we can implement motion blur attacks on the object (the bird) specifically, a natural corruption that could occur if birds move at speeds that are faster than the shutter speed. The aim is to be robust against all motion blur severity levels up to $M_{max} = 15$. To simulate motion blur, we apply a motion blur filter with a kernel of size $M$ on the segmented bird before we paste it onto the background image. We can change the severity level of the motion blur by increasing the kernel size of the filter. See Appendix D for concrete expressions of the motion blur kernel. Intuitively the worst attack should be the most severe blur, rendering a search over a range of severity superfluous. However, similar to rotations, this is not necessarily true in practice since the training loss on neural networks is generally nonconvex. Therefore, for an exact evaluation of the robust error at test time, we perform a full grid search over all kernel sizes in $[1, 2, \ldots, M_{max}]$. We refer to Figure 2d and Section D for an illustration of our motion blur attack. During training time, we perform an approximate search over kernels with sizes $2i$ for $i = 1, \ldots, M_{max}/2$.

**Adversarial illumination** As a second attack on the Waterbirds dataset, we consider adversarial illumination. The adversary can darken or brighten the bird without corrupting the background of the image. The attack aims to model images where the object at interest is hidden in shadows or placed against bright light. To compute the adversarial illumination attack, we modify the brightness of the segmented bird by adding a constant $a \in [-\epsilon_{\text{te}}, \epsilon_{\text{te}}]$ to all pixel values, before pasting the bird onto the background image. With an analogous argument as for the adversarial search for motion blur, the exact evaluation requires an actual search over the interval $[-\epsilon_{\text{te}}, \epsilon_{\text{te}}]$. We find the most adversarial lighting level, i.e. the value of $a$, by equidistantly partitioning the interval $[-\epsilon_{\text{te}}, \epsilon_{\text{te}}]$ in $K$ steps and performing a full list-search over all steps. See Figure 2c and Appendix D for an illustration of the adversarial illumination attack. We choose $K = 65, 33$ during test and training time respectively.

**Adversarial training** For all datasets and attacks, we run SGD until convergence on the *robust* cross-entropy loss 2. In each iteration, we search for an adversarial example as described above and

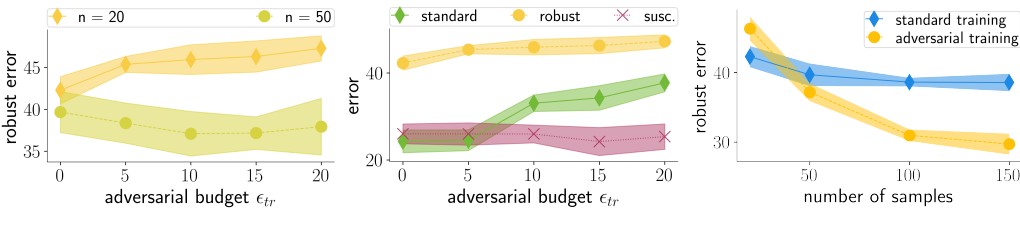

Figure 6: Experiments on the (subsampled) Waterbirds dataset using the motion blur attack. (a) Even though adversarial training hurts robust generalization for low sample size ($n = 20$), it helps for $n = 50$. (b) For $n = 20$, the decomposition of the robust error in standard error and susceptibility as a function of adversarial budget $\epsilon_{\mathrm{tr}}$. The increase in standard error is more severe than the drop in susceptibility, leading to a slight increase in robust error. (c) The robust error of standard and adversarial training on settings where the test error after standard training is small as a function of the number of samples. While adversarial training hurts for small sample sizes, it helps for larger sample sizes. For each experiment we plot the mean and standard deviation of the mean of independent experiments. For more experimental details see App. D.

update the weights using a gradient with respect to the resulting perturbed example (Goodfellow et al., 2015; Madry et al., 2018). For every experiment, we choose the learning rate and weight decay parameters that minimize the robust error on a hold-out dataset.

### 4.3 ADVERSARIAL TRAINING CAN HURT ROBUST GENERALIZATION

We now present our experimental results on the Waterbirds dataset for both motion blur and adversarial illumination attacks. First of all, Figure 5a and 6a show that the phenomenon characterized in the linear setting by Theorem 3.1 also occurs for directed attacks on the Waterbirds dataset: as we increase the adversarial training budget $\epsilon_{\mathrm{tr}}$ starting from zero (standard training), the robust error monotonically increases.

Furthermore, to gain intuition as described in Section 3.3, we also plot the robust error decomposition (Equation 7) consisting of the standard error and susceptibility in Figure 5b and 6b. Recall that we measure susceptibility as the fraction of data points in the test set for which the classifier predicts a different class under an adversarial attack. As in our linear example, we observe an increase in robust error despite a slight drop in susceptibility, because of the more severe increase in standard error. Moreover, Figures 1 and 6c show that analogous to our linear example, this phenomenon is specific to the low sample regime: for large sample size adversarial training outperforms standard training as expected. Note again that even the smallest sample size is large enough to yield a standard test error $< 10\%$ for standard training. Similar experiments for CIFAR-10 can be found in Appendix E. Finally, we empirically confirm in Appendix D.8 that our phenomenon is specific to directed attacks: for undirected attacks such as bounded $\ell_\infty-$and $\ell_2-$ball perturbations, adversarial training helps robust generalization also in the low sample size regime.

### 4.4 DISCUSSION

We now discuss how different algorithmic choices, motivated by related work, might affect how adversarial training hurts robust generalization.

**Catastrophic overfitting**   Often the worst-case perturbation during adversarial training is found using an approximate algorithm such as SGD. It is common belief that using the strongest attack (in the motion blur case, full grid search) during training also results in better robust generalization. In particular, the literature on catastrophic overfitting shows that weaker attacks during training lead to bad performance on stronger attacks during testing (Wong et al., 2020; Andriushchenko & Flammarion, 2020; Li et al., 2021). Our results suggests the opposite in the low sample size regime for directed attacks: the weaker the attack during training, the better adversarial training performs.

**Robust overfitting**   Recent work observes empirically (Rice et al., 2020) and theoretically (Sanyal et al., 2020; Donhauser et al., 2021), that perfectly minimizing the adversarial loss during training might in fact be suboptimal for robust generalization; that is, classical regularization techniques might lead to higher robust accuracy. This phenomenon is often referred to as robust overfitting. May the phenomenon be mitigated using standard regularization techniques? In Appendix D we shed light

on this question and show that adversarial training hurts robust generalization even when standard regularization methods such as early stopping are used.

## 5 RELATED WORK

**Robust and non-robust useful features**    In the words of Ilyas et al. (2019) and Springer et al. (2021) we can describe the intuition behind "our phenomenon" as follows: for directed attacks, all robust features become less useful, but adversarial training uses robust features more. In the small sample-size regime, $n < d - 1$ in particular, robust learning assigns too much weight on the robust (possibly non-useful) features that then dominate the non-robust (but useful)features. Even though they define these concepts, they don't make our statement, but show that adversarial training reduces the reliance on non-robust but possibly useful features.

**Small sample size and robustness**    A direct consequence of Theorem 3.1 is that in order to achieve the same robust error as standard training, adversarial training requires more samples. This statement might remind the reader of sample complexity results for robust generalization in Schmidt et al. (2018); Yin et al. (2019); Khim & Loh (2018). While those results compare sample complexity bounds for standard vs. robust error, our theorem statement compares two algorithms, standard vs. adversarial training, with respect to the robust error.

**Trade-off between standard and robust error**    Many papers observed that even though adversarial training decreases robust error compared to standard training, it may lead to an increase in standard test error Madry et al. (2018); Zhang et al. (2019). For example, Tsipras et al. (2019); Zhang et al. (2019); Javanmard et al. (2020); Dobriban et al. (2020); Chen et al. (2020) study settings where the Bayes optimal robust classifier is not equal to the Bayes optimal (standard) classifier (i.e. the perturbations are inconsistent or the dataset is non-separable). Raghunathan et al. (2020) study consistent perturbations, as in our paper, and prove that for small sample size, fitting adversarial examples can increase standard error even in the absence of noise. Empirically, Dong et al. (2021); Mendonça et al. (2022) show that for $\ell_p$-attacks low-quality data might be the main cause of the trade-off. While aforementioned works focus on the decrease in standard error, we prove that for directed attacks, in the small sample regime adversarial training may in fact increase *robust error*.

**Mitigation of the trade-off**    A long line of work has proposed procedures to mitigate the trade-off between robust and standard accuracy. For example Alayrac et al. (2019); Carmon et al. (2019); Zhai et al. (2019); Raghunathan et al. (2020) study robust self training, which leverages a large set of unlabelled data, while Lee et al. (2020); Lamb et al. (2019); Xu et al. (2020) use data augmentation by interpolation. Ding et al. (2020); Balaji et al. (2019); Cheng et al. (2020) on the other hand propose to use adaptive perturbation budgets $\epsilon_{\text{tr}}$ that vary across inputs. The intuition behind our theoretical analysis suggests that the standard mitigation procedures for imperceptible perturbations may not work for perceptible directed attacks, because all relevant features are non-robust. We leave a thorough empirical study as interesting future work.

## 6 SUMMARY AND FUTURE WORK

This paper aims to caution the practitioner against blindly following current widespread practices to increase the robust performance of machine learning models. Specifically, adversarial training is currently recognized to be one of the most effective defense mechanisms for $\ell_p$-perturbations, significantly outperforming robust performance of standard training. However, we prove that in the low sample size regime this common wisdom is not applicable for consistent directed attacks, which efficiently focus their attack budget to target the ground truth class information. In terms of follow-up work on directed attacks in the low sample regime, there are some concrete questions that would be interesting to explore. For example, as discussed in Section 5, it would be useful to test whether some methods to mitigate the standard accuracy vs. robustness trade-off would also relieve the perils of adversarial training for directed attacks. Further, we hypothesize that when few samples are available, one should avoid training with attacks that may heavily reduce class information, independently of the attacks at test time. If this hypothesis were confirmed, it would break with yet another general rule that the best defense perturbation type should always match the attack during evaluation.

ACKNOWLEDGEMENT

Supported by the Hasler Foundation grant number 21050.

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

## A   THEORETICAL STATEMENTS FOR THE LINEAR MODEL

Before we present the proof of the theorem, we introduce two lemmas are of separate interest that are used throughout the proof of Theorem 1. Recall that the definition of the (standard normalized) maximum-$\ell_2$-margin solution (max-margin solution in short) of a dataset $D = \{(x_i, y_i)\}_{i=1}^n$ corresponds to

$$\widehat{\theta} := \arg\max_{\|\theta\|_2 \leq 1} \min_{i \in [n]} y_i \theta^\top x_i, \tag{10}$$

by simply setting $\epsilon_{\text{tr}} = 0$ in Equation 4. The $\ell_2$-margin of $\widehat{\theta}$ then reads $\min_{i \in [n]} y_i \widehat{\theta}^\top x_i$. Furthermore for a dataset $D = \{(x_i, y_i)\}_{i=1}^n$ we refer to the induced dataset $\widetilde{D}$ as the dataset with covariate vectors stripped of the first element, i.e.

$$\widetilde{D} = \{(\tilde{x}_i, y_i)\}_{i=1}^n := \{((x_i)_{[2:d]}, y_i)\}_{i=1}^n, \tag{11}$$

where $(x_i)_{[2:d]}$ refers to the last $d - 1$ elements of the vector $x_i$. Furthermore, remember that for any vector $z$, $z_{[j]}$ refers to the $j$-th element of $z$ and $e_j$ denotes the $j$-th canonical basis vector. Further, recall the distribution $\mathbb{P}_r$ as defined in Section 3.1: the label $y \in \{+1, -1\}$ is drawn with equal probability and the covariate vector is sampled as $x = [y\frac{r}{2}, \tilde{x}]$ where $\tilde{x} \in \mathbb{R}^{d-1}$ is a random vector drawn from a standard normal distribution, i.e. $\tilde{x} \sim \mathcal{N}(0, \sigma^2 I_{d-1})$. We generally allow $r$, used to sample the training data, to differ from $r_{\text{test}}$, which is used during test time.

The following lemma derives a closed-form expression for the normalized max-margin solution for any dataset with fixed separation $r$ in the signal component, and that is linearly separable in the last $d - 1$ coordinates with margin $\tilde{\gamma}$.

**Lemma A.1.** *Let $D = \{(x_i, y_i)\}_{i=1}^n$ be a dataset that consists of points $(x, y) \in \mathbb{R}^d \times \{\pm 1\}$ and $x_{[1]} = y\frac{r}{2}$, i.e. the covariates $x_i$ are deterministic in their first coordinate given $y_i$ with separation distance $r$. Furthermore, let the induced dataset $\widetilde{D}$ also be linearly separable by the normalized max-$\ell_2$-margin solution $\tilde{\theta}$ with an $\ell_2$-margin $\tilde{\gamma}$. Then, the normalized max-margin solution of the original dataset $D$ is given by*

$$\widehat{\theta} = \frac{1}{\sqrt{r^2 + 4\tilde{\gamma}^2}} \left[ r, 2\tilde{\gamma}\tilde{\theta} \right]. \tag{12}$$

*Further, the standard accuracy of $\widehat{\theta}$ for data drawn from $\mathbb{P}_{r_{\text{test}}}$ reads*

$$\mathbb{P}_{r_{\text{test}}}(Y\widehat{\theta}^\top X > 0) = \Phi\left( \frac{r \, r_{\text{test}}}{4\sigma \, \tilde{\gamma}} \right). \tag{13}$$

The proof can be found in Section A.3. The next lemma provides high probability upper and lower bounds for the margin $\tilde{\gamma}$ of $\widetilde{D}$ when $\tilde{x}_i$ are drawn from the normal distribution.

**Lemma A.2.** *Let $\widetilde{D} = \{(\tilde{x}_i, y_i)\}_{i=1}^n$ be a random dataset where $y_i \in \{\pm 1\}$ are equally distributed and $\tilde{x}_i \sim \mathcal{N}(0, \sigma I_{d-1})$ for all $i$, and $\tilde{\gamma}$ is the maximum $\ell_2$ margin that can be written as*

$$\tilde{\gamma} = \max_{\|\tilde{\theta}\|_2 \leq 1} \min_{i \in [n]} y_i \tilde{\theta}^\top \tilde{x}_i.$$

*Then, for any $t \geq 0$, with probability greater than $1 - 2e^{-\frac{t^2}{2}}$, we have $\tilde{\gamma}_{\min}(t) \leq \tilde{\gamma} \leq \tilde{\gamma}_{\max}(t)$ where*

$$\tilde{\gamma}_{\max}(t) = \sigma\left( \sqrt{\frac{d-1}{n}} + 1 + \frac{t}{\sqrt{n}} \right), \quad \tilde{\gamma}_{\min}(t) = \sigma\left( \sqrt{\frac{d-1}{n}} - 1 - \frac{t}{\sqrt{n}} \right).$$

### A.1   PROOF OF THEOREM 3.1

Given a dataset $D = \{(x_i, y_i)\}$ drawn from $\mathbb{P}_r$, it is easy to see that the (normalized) $\epsilon_{\text{tr}}$-robust max-margin solution 4 of $D$ with respect to signal-attacking perturbations $T(\epsilon_{\text{tr}}; x_i)$ as defined in Equation 3, can be written as

$$\begin{aligned}
\widehat{\theta}^{\epsilon_{\text{tr}}} &= \arg\max_{\|\theta\|_2 \leq 1} \min_{i \in [n], x_i' \in T(x_i; \epsilon_{\text{tr}})} y_i \theta^\top x_i' \\
&= \arg\max_{\|\theta\|_2 \leq 1} \min_{i \in [n], |\beta| \leq \epsilon_{\text{tr}}} y_i \theta^\top (x_i + \beta e_1) \\
&= \arg\max_{\|\theta\|_2 \leq 1} \min_{i \in [n]} y_i \theta^\top (x_i - y_i \epsilon_{\text{tr}} \operatorname{sign}(\theta_{[1]}) e_1).
\end{aligned}$$

Note that by definition, it is equivalent to the (standard normalized) max-margin solution $\widehat{\theta}$ of the shifted dataset $D_{\epsilon_{\text{tr}}} = \{(x_i - y_i\epsilon_{\text{tr}}\,\text{sign}(\theta_{[1]})e_1, y_i)\}_{i=1}^n$. Since $D_{\epsilon_{\text{tr}}}$ satisfies the assumptions of Lemma A.1, it then follows directly that the normalized $\epsilon_{\text{tr}}$-robust max-margin solution reads

$$\widehat{\theta}^{\epsilon_{\text{tr}}} = \frac{1}{\sqrt{(r - 2\epsilon_{\text{tr}})^2 + 4\tilde{\gamma}^2}} \left[ r - 2\epsilon_{\text{tr}}, 2\tilde{\gamma}\tilde{\theta} \right], \tag{14}$$

by replacing $r$ by $r - 2\epsilon_{\text{tr}}$ in Equation 12. Similar to above, $\tilde{\theta} \in R^{d-1}$ is the (standard normalized) max-margin solution of $\{(\tilde{x}_i, y_i)\}_{i=1}^n$ and $\tilde{\gamma}$ the corresponding margin.

**Proof of 1.** We can now compute the $\epsilon_{\text{te}}$-robust accuracy of the $\epsilon_{\text{tr}}$-robust max-margin estimator $\widehat{\theta}^{\epsilon_{\text{tr}}}$ for a given dataset $D$ as a function of $\tilde{\gamma}$. Note that in the expression of $\widehat{\theta}^{\epsilon_{\text{tr}}}$, all values are fixed for a fixed dataset, while $0 \leq \epsilon_{\text{tr}} \leq r - 2\tilde{\gamma}_{\max}$ can be chosen. First note that for a test distribution $\mathbb{P}_r$, the $\epsilon_{\text{te}}$-robust accuracy, defined as one minus the robust error (Equation 1), for a classifier associated with a vector $\theta$, can be written as

$$\text{Acc}(\theta; \epsilon_{\text{te}}) = \mathbb{E}_{X,Y\sim\mathbb{P}_r} \left[ \mathbb{I}\{ \min_{x' \in T(X;\epsilon_{\text{te}})} Y\theta^\top x' > 0 \} \right] \tag{15}$$

$$= \mathbb{E}_{X,Y\sim\mathbb{P}_r} \left[ \mathbb{I}\{Y\theta^\top X - \epsilon_{\text{te}}\theta_{[1]} > 0\} \right] = \mathbb{E}_{X,Y\sim\mathbb{P}_r} \left[ \mathbb{I}\{Y\theta^\top (X - Y\epsilon_{\text{te}}\,\text{sign}(\theta_{[1]})e_1) > 0\} \right]$$

Now, recall that by Equation 14 and the assumption in the theorem, we have $r - 2\epsilon_{\text{tr}} > 0$, so that $\text{sign}(\widehat{\theta}^{\epsilon_{\text{tr}}}) = 1$. Further, using the definition of the $T(\epsilon_{\text{tr}}; x)$ in Equation 3 and by definition of the distribution $\mathbb{P}_r$, we have $X_{[1]} = Y\frac{r}{2}$. Plugging into Equation 15 then yields

$$\text{Acc}(\widehat{\theta}^{\epsilon_{\text{tr}}}; \epsilon_{\text{te}}) = \mathbb{E}_{X,Y\sim\mathbb{P}_r} \left[ \mathbb{I}\{Y\widehat{\theta}^{\epsilon_{\text{tr}}\top}(X - Y\epsilon_{\text{te}}e_1) > 0\} \right]$$

$$= \mathbb{E}_{X,Y\sim\mathbb{P}_r} \left[ \mathbb{I}\{Y\widehat{\theta}^{\epsilon_{\text{tr}}\top}(X_{-1} + Y\left(\frac{r}{2} - \epsilon_{\text{te}}\right)e_1) > 0\} \right]$$

$$= \mathbb{P}_{r-2\epsilon_{\text{te}}}(Y\widehat{\theta}^{\epsilon_{\text{tr}}\top}X > 0)$$

where $X_{-1}$ is a shorthand for the random vector $X_{-1} = (0; X_{[2]}, \ldots, X_{[d]})$. The assumptions in Lemma A.1 ($D_{\epsilon_{\text{tr}}}$ is linearly separable) are satisfied whenever the $n < d - 1$ samples are distinct, i.e. with probability one. Hence applying Lemma A.1 with $r_{\text{test}} = r - 2\epsilon_{\text{te}}$ and $r = r - 2\epsilon_{\text{tr}}$ yields

$$\text{Acc}(\widehat{\theta}^{\epsilon_{\text{tr}}}; \epsilon_{\text{te}}) = \Phi\left( \frac{r(r - 2\epsilon_{\text{te}})}{4\sigma\tilde{\gamma}} - \epsilon_{\text{tr}}\frac{r - 2\epsilon_{\text{te}}}{2\sigma\tilde{\gamma}} \right). \tag{16}$$

Theorem statement a) then follows by noting that $\Phi$ is a monotically decreasing function in $\epsilon_{\text{tr}}$. The expression for the robust error then follows by noting that $1 - \Phi(-z) = \Phi(z)$ for any $z \in \mathbb{R}$ and defining

$$\tilde{\varphi} = \frac{\sigma\tilde{\gamma}}{r/2 - \epsilon_{\text{te}}}. \tag{17}$$

**Proof of 2.** First define $\varphi_{\min}, \varphi_{\max}$ using $\tilde{\gamma}_{\min}, \tilde{\gamma}_{\max}$ as in Equation 17. Then we have by Equation 16

$$\text{Err}(\widehat{\theta}^{\epsilon_{\text{tr}}}; \epsilon_{\text{te}}) - \text{Err}(\widehat{\theta}^0; \epsilon_{\text{te}}) = \text{Acc}(\widehat{\theta}^0; \epsilon_{\text{te}}) - \text{Acc}(\widehat{\theta}^{\epsilon_{\text{tr}}}; \epsilon_{\text{te}})$$

$$= \Phi\left(\frac{r/2}{\tilde{\varphi}}\right) - \Phi\left(\frac{r/2 - \epsilon_{\text{tr}}}{\tilde{\varphi}}\right)$$

$$= \int_{r/2-\epsilon_{\text{tr}}}^{r/2} \frac{1}{\sqrt{2\pi}\tilde{\varphi}} \mathbb{E}^{-\frac{x^2}{\tilde{\varphi}^2}} dx$$

By plugging in $t = \sqrt{\frac{2\log 2/\delta}{n}}$ in Lemma A.2, we obtain that with probability at least $1 - \delta$ we have

$$\tilde{\gamma}_{\min} := \sigma\left[\sqrt{\frac{d-1}{n}} - \left(1 + \sqrt{\frac{2\log(2/\delta)}{n}}\right)\right] \leq \tilde{\gamma} \leq \sigma\left[\sqrt{\frac{d-1}{n}} + \left(1 + \sqrt{\frac{2\log(2/\delta)}{n}}\right)\right] =: \tilde{\gamma}_{\max}$$

and equivalently $\varphi_{\min} \leq \tilde{\varphi} \leq \varphi_{\max}$.

Now note the general fact that for all $\tilde{\varphi} \leq \sqrt{2}x$ the density function $f(\tilde{\varphi}; x) = \frac{1}{\sqrt{2\pi}\tilde{\varphi}} \mathbb{E}^{-\frac{x^2}{\tilde{\varphi}^2}}$ is monotonically increasing in $\tilde{\varphi}$.

By assumption of the theorem, $\tilde{\varphi} \leq \sqrt{2}(r/2 - \epsilon_{\mathrm{tr}})(r/2 - \epsilon_{\mathrm{te}})$ so that $f(\tilde{\varphi}; x) \geq f(\varphi_{\min}; x)$ for all $x \in [r/2 - \epsilon_{\mathrm{tr}}, r/2]$ and therefore

$$\int_{r/2-\epsilon_{\mathrm{tr}}}^{r/2} \frac{1}{\sqrt{2\pi}\tilde{\varphi}} \mathbb{E}^{-\frac{x^2}{\tilde{\varphi}^2}} dx \geq \int_{r/2-\epsilon_{\mathrm{tr}}}^{r/2} \frac{1}{\sqrt{2\pi}\varphi_{\min}} \mathbb{E}^{-\frac{x^2}{\tilde{\varphi}^2}} dx = \Phi\left(\frac{r/2}{\varphi_{\min}}\right) - \Phi\left(\frac{r/2 - \epsilon_{\mathrm{tr}}}{\varphi_{\min}}\right).$$

and the statement is proved.

## A.2 PROOF OF COROLLARY 3.2

We now show that Theorem 3.1 also holds for $\ell_1$-ball perturbations with at most radius $\epsilon$. Following similar steps as in Equation 14, the $\epsilon_{\mathrm{tr}}$-robust max-margin solution for $\ell_1$-perturbations can be written as

$$\widehat{\theta}^{\epsilon_{\mathrm{tr}}} := \arg\max_{\|\theta\|_2 \leq 1} \min_{i \in [n]} y_i \theta^\top (x_i - y_i \epsilon_{\mathrm{tr}} \operatorname{sign}(\theta_{[j^\star(\theta)]}) e_{j^\star(\theta)}) \tag{18}$$

where $j^\star(\theta) := \arg\max_j |\theta_j|$ is the index of the maximum absolute value of $\theta$. We now prove by contradiction that the robust max-margin solution for this perturbation set 9 is equivalent to the solution 14 for the perturbation set 3. We start by assuming that $\widehat{\theta}^{\epsilon_{\mathrm{tr}}}$ does not solve Equation 14, which is equivalent to assuming $1 \notin j^\star(\widehat{\theta}^{\epsilon_{\mathrm{tr}}})$ by definition. We now show how this assumption leads to a contradiction.

Define the shorthand $j^\star := j^\star(\widehat{\theta}^{\epsilon_{\mathrm{tr}}}) - 1$. Since $\widehat{\theta}^{\epsilon_{\mathrm{tr}}}$ is the solution of 18, by definition, we have that $\widehat{\theta}^{\epsilon_{\mathrm{tr}}}$ is also the max-margin solution of the shifted dataset $D_{\epsilon_{\mathrm{tr}}} := (x_i - y_i \epsilon_{\mathrm{tr}} \operatorname{sign}(\theta_{[j^\star+1]}) e_{j^\star+1}, y_i)$. Further, note that by the assumption that $1 \notin j^\star(\widehat{\theta}^{\epsilon_{\mathrm{tr}}})$, this dataset $D_{\epsilon_{\mathrm{tr}}}$ consists of input vectors $x_i = (y_i \frac{r}{2}, \tilde{x}_i - y_i \epsilon_{\mathrm{tr}} \operatorname{sign}(\theta_{[j^\star+1]}) e_{j^\star+1})$. Hence via Lemma A.1, $\widehat{\theta}^{\epsilon_{\mathrm{tr}}}$ can be written as

$$\widehat{\theta}^{\epsilon_{\mathrm{tr}}} = \frac{1}{\sqrt{r^2 - 4(\tilde{\gamma}^{\epsilon_{\mathrm{tr}}})^2}} [r, 2\tilde{\gamma}^{\epsilon_{\mathrm{tr}}} \tilde{\theta}^{\epsilon_{\mathrm{tr}}}], \tag{19}$$

where $\tilde{\theta}^{\epsilon_{\mathrm{tr}}}$ is the normalized max-margin solution of $\widetilde{D} := (\tilde{x}_i - y_i \epsilon_{\mathrm{tr}} \operatorname{sign}(\theta_{[j^\star]}) e_{j^\star}, y_i)$.

We now characterize $\tilde{\theta}^{\epsilon_{\mathrm{tr}}}$. Note that by assumption, $j^\star = j^\star(\tilde{\theta}^{\epsilon_{\mathrm{tr}}}) = \arg\max_j |\tilde{\theta}_{[j]}^{\epsilon_{\mathrm{tr}}}|$. Hence, the normalized max-margin solution $\tilde{\theta}^{\epsilon_{\mathrm{tr}}}$ is the solution of

$$\tilde{\theta}^{\epsilon_{\mathrm{tr}}} := \arg\max_{\|\tilde{\theta}\|_2 \leq 1} \min_{i \in [n]} y_i \tilde{\theta}^\top \tilde{x}_i - \epsilon_{\mathrm{tr}} |\tilde{\theta}_{[j^\star]}| \tag{20}$$

Observe that the minimum margin of this estimator $\tilde{\gamma}^{\epsilon_{\mathrm{tr}}} = \min_{i \in [n]} y_i (\tilde{\theta}^{\epsilon_{\mathrm{tr}}})^\top \tilde{x}_i - \epsilon_{\mathrm{tr}} |\tilde{\theta}_{[j^\star]}^{\epsilon_{\mathrm{tr}}}|$ decreases with $\epsilon_{\mathrm{tr}}$ as the problem becomes harder $\tilde{\gamma}^{\epsilon_{\mathrm{tr}}} \leq \tilde{\gamma}$, where the latter is equivalent to the margin of $\tilde{\theta}^{\epsilon_{\mathrm{tr}}}$ for $\epsilon_{\mathrm{tr}} = 0$. Since $r > 2\tilde{\gamma}_{\max}$ by assumption in the Theorem, by Lemma A.2 with probability at least $1 - 2\mathbb{E}^{-\frac{\alpha^2(d-1)}{n}}$, we then have that $r > 2\tilde{\gamma} \geq 2\tilde{\gamma}^{\epsilon_{\mathrm{tr}}}$. Given the closed form of $\widehat{\theta}^{\epsilon_{\mathrm{tr}}}$ in Equation 19, it directly follows that $\widehat{\theta}_{[1]}^{\epsilon_{\mathrm{tr}}} = r > 2\tilde{\gamma}^{\epsilon_{\mathrm{tr}}} \|\tilde{\theta}^{\epsilon_{\mathrm{tr}}}\|_2 = \|\widehat{\theta}_{[2:d]}^{\epsilon_{\mathrm{tr}}}\|_2$ and hence $1 \in j^\star(\widehat{\theta}^{\epsilon_{\mathrm{tr}}})$. This contradicts the original assumption $1 \notin j^\star(\widehat{\theta}^{\epsilon_{\mathrm{tr}}})$ and hence we established that $\widehat{\theta}^{\epsilon_{\mathrm{tr}}}$ for the $\ell_1$-perturbation set 9 has the same closed form 14 as for the perturbation set 3.

The final statement is proved by using the analogous steps as in the proof of 1. and 2. to obtain the closed form of the robust accuracy of $\widehat{\theta}^{\epsilon_{\mathrm{tr}}}$.

## A.3 PROOF OF LEMMA A.1

We start by proving that $\widehat{\theta}$ is of the form

$$\widehat{\theta} = \left[a_1, a_2 \tilde{\theta}\right], \tag{21}$$

for $a_1, a_2 > 0$. Denote by $\mathcal{H}(\theta)$ the plane through the origin with normal $\theta$. We define $d\left((x, y), \mathcal{H}(\theta)\right)$ as the signed euclidean distance from the point $(x, y) \in D \sim \mathbb{P}_r$ to the plane $\mathcal{H}(\theta)$. The signed

euclidean distance is the defined as the euclidean distance from x to the plane if the point $(x, y)$ is correctly predicted by $\theta$, and the negative euclidean distance from $x$ to the plane otherwise. We rewrite the definition of the max $l_2$-margin classifier. It is the classifier induced by the normalized vector $\widehat{\theta}$, such that

$$\max_{\theta \in \mathbb{R}^d} \min_{(x,y) \in D} d\left((x, y), \mathcal{H}(\theta)\right) = \min_{(x,y) \in D} d\left((x, y), \mathcal{H}(\widehat{\theta})\right).$$

We use that $D$ is deterministic in its first coordinate and get

$$\max_{\theta} \min_{(x,y) \in D} d\left((x, y), \mathcal{H}(\theta)\right) = \max_{\theta} \min_{(x,y) \in D} y(\theta_{[1]} x_{[1]} + \tilde{\theta}^\top \tilde{x})$$

$$= \max_{\theta} \theta_1 \frac{r}{2} + \min_{(x,y) \in D} y \tilde{\theta}^\top \tilde{x}.$$

Because $r > 0$, the maximum over all $\theta$ has $\widehat{\theta}_{[1]} \geq 0$. Take any $a > 0$ such that $\|\tilde{\theta}\|_2 = a$. By definition the max $l_2$-margin classifier, $\tilde{\theta}$, maximizes $\min_{(x,y) \in D} d\left((x, y), \mathcal{H}(\theta)\right)$. Therefore, $\widehat{\theta}$ is of the form of Equation 21.

Note that all classifiers induced by vectors of the form of Equation 21 classify $D$ correctly. Next, we aim to find expressions for $a_1$ and $a_2$ such that Equation 21 is the normalized max $l_2$-margin classifier. The distance from any $x \in D$ to $\mathcal{H}(\widehat{\theta})$ is

$$d\left(x, \mathcal{H}(\widehat{\theta})\right) = \left| a_1 x_{[1]} + a_2 \tilde{\theta}^\top \tilde{x} \right|.$$

Using that $x_{[1]} = y \frac{r}{2}$ and that the second term equals $a_2 d\left(x, \mathcal{H}(\tilde{\theta})\right)$, we get

$$d\left(x, \mathcal{H}(\widehat{\theta})\right) = \left| a_1 \frac{r}{2} + a_2 d\left(x, \mathcal{H}(\tilde{\theta})\right) \right| = a_1 \frac{r}{2} + \sqrt{1 - a_1^2} d\left(x, \mathcal{H}(\tilde{\theta})\right). \tag{22}$$

Let $(\tilde{x}, y) \in \widetilde{D}$ be the point closest in Euclidean distance to $\tilde{\theta}$. This point is also the closest point in Euclidean distance to $\mathcal{H}(\widehat{\theta})$, because by Equation 22 $d\left(x, \mathcal{H}(\widehat{\theta})\right)$ is strictly decreasing for decreasing $d\left(x, \mathcal{H}(\tilde{\theta})\right)$. We maximize the minimum margin $d\left(x, \mathcal{H}(\widehat{\theta})\right)$ with respect to $a_1$. Define the vectors $a = [a_1, a_2]$ and $v = \left[\frac{r}{2}, d\left(x, \mathcal{H}(\tilde{\theta})\right)\right]$. We find using the dual norm that

$$a = \frac{v}{\|v\|_2}.$$

Plugging the expression of $a$ into Equation 21 yields that $\widehat{\theta}$ is given by

$$\widehat{\theta} = \frac{1}{\sqrt{r^2 + 4\tilde{\gamma}^2}} \left[r, 2\tilde{\gamma}\tilde{\theta}\right].$$

For the second part of the lemma we first decompose

$$\mathbb{P}_{r_{\text{test}}}(Y\widehat{\theta}^\top X > 0) = \frac{1}{2}\mathbb{P}_{r_{\text{test}}}\left[\widehat{\theta}^\top X > 0 \mid Y = 1\right] + \frac{1}{2}\mathbb{P}_{r_{\text{test}}}\left[\widehat{\theta}^\top X < 0 \mid Y = -1\right]$$

We can further write

$$\mathbb{P}_{r_{\text{test}}}\left[\widehat{\theta}^\top X > 0 \mid Y = 1\right] = \mathbb{P}_{r_{\text{test}}}\left[\sum_{i=2}^{d} \widehat{\theta}_{[i]} X_{[i]} > -\widehat{\theta}_{[1]} X_{[1]} \mid Y = 1\right] \tag{23}$$

$$= \mathbb{P}_{r_{\text{test}}}\left[2\tilde{\gamma} \sum_{i=1}^{d-1} \tilde{\theta}_{[i]} X_{[i]} > -r \frac{r_{\text{test}}}{2} \mid Y = 1\right]$$

$$= 1 - \Phi\left(-\frac{r\, r_{\text{test}}}{4\sigma\tilde{\gamma}}\right) = \Phi\left(\frac{r\, r_{\text{test}}}{4\sigma\tilde{\gamma}}\right)$$

where $\Phi$ is the cumulative distribution function. The second equality follows by multiplying by the normalization constant on both sides and the third equality is due to the fact that $\sum_{i=1}^{d-1} \tilde{\theta}_{[i]} X_{[i]}$ is

a zero-mean Gaussian with variance $\sigma^2 \|\tilde{\theta}\|_2^2 = \sigma^2$ since $\tilde{\theta}$ is normalized. Correspondingly we can write

$$\mathbb{P}_{r_{\text{test}}} \left[ \widehat{\theta}^\top X < 0 \mid Y = -1 \right] = \mathbb{P}_{r_{\text{test}}} \left[ 2\tilde{\gamma} \sum_{i=1}^{d-1} \tilde{\theta}_{[i]} X_{[i]} < -r \left( -\frac{r_{\text{test}}}{2} \right) \mid Y = -1 \right] = \Phi \left( \frac{r\, r_{\text{test}}}{4\sigma\tilde{\gamma}} \right) \tag{24}$$

so that we can combine 23 and 23 and 24 to obtain $\mathbb{P}_{r_{\text{test}}} (Y\widehat{\theta}^\top X > 0) = \Phi \left( \frac{r\, r_{\text{test}}}{4\sigma\tilde{\gamma}} \right)$. This concludes the proof of the lemma.

## A.4 PROOF OF LEMMA A.2

The proof plan is as follows. We start from the definition of the max $\ell_2$-margin of a dataset. Then, we rewrite the max $\ell_2$-margin as an expression that includes a random matrix with independent standard normal entries. This allows us to prove the upper and lower bounds for the max-$\ell_2$-margin in Sections A.4.1 and A.4.2 respectively, using non-asymptotic estimates on the singular values of Gaussian random matrices.

Given the dataset $\widetilde{D} = \{(\tilde{x}_i, y_i)\}_{i=1}^n$, we define the random matrix

$$X = \begin{pmatrix} \tilde{x}_1^\top \\ \tilde{x}_2^\top \\ \dots \\ \tilde{x}_n^\top \end{pmatrix}. \tag{25}$$

where $\tilde{x}_i \sim \mathcal{N}(0, \sigma I_{d-1})$. Let $\mathcal{V}$ be the class of all perfect predictors of $\widetilde{D}$. For a matrix $A$ and vector $b$ we also denote by $|Ab|$ the vector whose entries correspond to the absolute values of the entries of $Ab$. Then, by definition

$$\tilde{\gamma} = \max_{v \in \mathcal{V}, \|v\|_2 = 1} \min_{j \in [n]} |Xv|_{[j]} = \max_{v \in \mathcal{V}, \|v\|_2 = 1} \min_{j \in [n]} \sigma |Qv|_{[j]}, \tag{26}$$

where $Q = \frac{1}{\sigma} X$ is the scaled data matrix.

In the sequel we will use the operator norm of a matrix $A \in \mathbb{R}^{n \times d - 1}$.

$$\|A\|_2 = \sup_{v \in \mathbb{R}^{d-1} | \|v\|_2 = 1} \|Av\|_2$$

and denote the maximum singular value of a matrix $A$ as $s_{\max}(A)$ and the minimum singular value as $s_{\min}(A)$.

### A.4.1 UPPER BOUND

Given the maximality of the operator norm and since the minimum entry of the vector $|Qv|$ must be smaller than $\frac{\|Q\|_2}{\sqrt{n}}$, we can upper bound $\tilde{\gamma}$ by

$$\tilde{\gamma} \le \sigma \frac{1}{\sqrt{n}} \|Q\|_2.$$

Taking the expectation on both sides with respect to the draw of $\widetilde{D}$ and noting $\|Q\|_2 \le s_{\max}(Q)$, it follows from Corollary 5.35 of Vershynin (2010) that for all $t \ge 0$:

$$\mathbb{P} \left[ \sqrt{d-1} + \sqrt{n} + t \ge s_{\max}(Q) \right] \ge 1 - 2e^{-\frac{t^2}{2}}.$$

Therefore, with a probability greater than $1 - 2e^{-\frac{t^2}{2}}$,

$$\tilde{\gamma} \le \sigma \left( 1 + \frac{t + \sqrt{d-1}}{\sqrt{n}} \right).$$

### A.4.2 LOWER BOUND

By the definition in Equation 26, if we find a vector $v \in \mathcal{V}$ with $\|v\|_2 = 1$ such that for an $a > 0$, it holds that $\min_{j \in n} \sigma |Xv|_{[j]} > a$, then $\tilde{\gamma} > a$.

Recall the definition of the max-$\ell_2$-margin as in Equation 25. As $n < d - 1$, the random matrix $Q$ is a wide matrix, i.e. there are more columns than rows and therefore the minimal singular value is $0$. Furthermore, $Q$ has rank $n$ almost surely and hence for all $c > 0$, there exists a $v \in \mathbb{R}^{d-1}$ such that

$$\sigma Q v = 1_{\{n\}} c > 0, \tag{27}$$

where $1_{\{n\}}$ denotes the all ones vector of dimension $n$. The smallest non-zero singular value of $Q$, $s_{\text{min, nonzero}}(Q)$, equals the smallest non-zero singular value of its transpose $Q^\top$. Therefore, there also exists a $v \in \mathcal{V}$ with $\|v\|_2 = 1$ such that

$$\tilde{\gamma} \geq \min_{j \in [n]} \sigma |Qv|_{[j]} \geq \sigma s_{\text{min,nonzeros}} \left( Q^\top \right) \frac{1}{\sqrt{n}}, \tag{28}$$

where we used the fact that any vector $v$ in the span of non-zero eigenvectors satisfies $\|Qv\|_2 \geq s_{\text{min, nonzeros}}(Q)$ and the existence of a solution $v$ for any right-hand side as in Equation 27. Taking the expectation on both sides, Corollary 5.35 of Vershynin (2010) yields that with a probability greater than $1 - 2e^{-\frac{t^2}{2}}, t \geq 0$ we have

$$\tilde{\gamma} \geq \sigma \left( \frac{\sqrt{d-1} - t}{\sqrt{n}} - 1 \right). \tag{29}$$

## B BOUNDS ON THE SUSCEPTIBILITY SCORE

In Theorem 3.1, we give non-asymptotic bounds on the robust and standard error of a linear classifier trained with adversarial logistic regression. Moreover, we use the robust error decomposition in susceptibility and standard error to gain intuition about how adversarial training may hurt robust generalization. In this section, we complete the result of Theorem 3.1 by also deriving non-asymptotic bounds on the susceptibility score of the max $\ell_2$-margin classifier.

Using the results in Appendix A, we can prove following Corollary B.1, which gives non asymptotic bounds on the susceptibility score.

**Corollary B.1.** *Assume $d - 1 > n$. For the $\epsilon_{te}$-susceptibility on test samples from $\mathbb{P}_r$ with $2\epsilon_{te} < r$ and perturbation sets in Equation equation 3 and equation 9 the following holds:*

*For $\epsilon_{tr} < \frac{r}{2} - \tilde{\gamma}_{\max}$, with probability at least $1 - 2\mathbb{E}^{-\frac{\alpha^2(d-1)}{2}}$ for any $0 < \alpha < 1$, over the draw of a dataset $D$ with $n$ samples from $\mathbb{P}_r$, the $\epsilon_{te}$-susceptibility is upper and lower bounded by*

$$\begin{aligned}
Susc(\widehat{\theta}^{\epsilon_{tr}}; \epsilon_{te}) &\leq \Phi \left( \frac{(r - 2\epsilon_{tr})(\epsilon_{te} - \frac{r}{2})}{2\tilde{\gamma}_{\max}\sigma} \right) - \Phi \left( \frac{(r - 2\epsilon_{tr})(-\epsilon_{te} - \frac{r}{2})}{2\tilde{\gamma}_{\min}\sigma} \right) \\
Susc(\widehat{\theta}^{\epsilon_{tr}}; \epsilon_{te}) &\geq \Phi \left( \frac{(r - 2\epsilon_{tr})(\epsilon_{te} - \frac{r}{2})}{2\tilde{\gamma}_{\min}\sigma} \right) - \Phi \left( \frac{(r - 2\epsilon_{tr})(-\epsilon_{te} - \frac{r}{2})}{2\tilde{\gamma}_{\max}\sigma} \right)
\end{aligned} \tag{30}$$

We give the proof in Subsection B.1. Observe that the bounds on the susceptibility score in Corollary B.1 consist of two terms each, where the second term decreases with $\epsilon_{tr}$, but the first term increases. We recognise following two regimes: the max $\ell_2$-margin classifier is close to the ground truth $e_1$ or not. Clearly, the ground truth classifier has zero susceptibility and hence classifiers close to the ground truth also have low susceptibility. On the other hand, if the max $l_2$-margin classifier is not close to the ground truth, then putting less weight on the first coordinate increases invariance to the perturbations along the first direction. Recall that by Lemma A.1, increasing $\epsilon_{tr}$, decreases the weight on the first coordinate of the max $\ell_2$-margin classifier. Furthermore, in the low sample size regime, we are likely not close to the ground truth. Therefore, the regime where the susceptibility decreases with increasing $\epsilon_{tr}$ dominates in the low sample size regime.

To confirm the result of Corollary B.1, we plot the mean and standard deviation of the susceptibility score of 5 independent experiments. The results are depicted in Figure 7. We see that for low standard

error, when the classifier is reasonably close to the optimal classifier, the susceptibility increases slightly with increasing adversarial budget. However, increasing the adversarial training budget, $\epsilon_{\text{tr}}$, further, causes the susceptibility score to drop greatly. Hence, we can recognize both regimes and validate that, indeed, the second regime dominates in the low sample size setting.

### B.1 PROOF OF COROLLARY B.1

We proof the statement by bounding the robustness of a linear classifier. Recall that the robustness of a classifier is the probability that a classifier does not change its prediction under an adversarial attack. The susceptibility score is then given by

$$\text{Susc}(\widehat{\theta}^{\epsilon_{\text{tr}}}; \epsilon_{\text{te}}) = 1 - \text{Rob}(\widehat{\theta}^{\epsilon_{\text{tr}}}; \epsilon_{\text{te}}). \tag{31}$$

The proof idea is as follows: since the perturbations are along the first basis direction, $e_1$, we compute the distance from the robust $l_2$-max margin $\widehat{\theta}^{\epsilon_{\text{tr}}}$ to a point $(X, Y) \sim \mathbb{P}$. Then, we note that the robustness of $\widehat{\theta}^{\epsilon_{\text{tr}}}$ is given by the probability that the distance along $e_1$, from $X$ to the decision plane induced by $\widehat{\theta}^{\epsilon_{\text{tr}}}$ is greater then $\epsilon_{\text{te}}$. Lastly, we use the non-asymptotic bounds of Lemma A.2.

Recall, by Lemma A.1, the max $l_2$-margin classifier is of the form of

$$\widehat{\theta}^{\epsilon_{\text{tr}}} = \frac{1}{\sqrt{(r - 2\epsilon_{\text{tr}})^2 + 4\tilde{\gamma}^2}} \left[ r - 2\epsilon_{\text{tr}}, 2\tilde{\gamma}\tilde{\theta} \right]. \tag{32}$$

Let $(X, Y) \sim \mathbb{P}$. The distance along $e_1$ from $X$ to the decision plane induced by $\widehat{\theta}^{\epsilon_{\text{tr}}}$, $\mathcal{H}(\widehat{\theta}^{\epsilon_{\text{tr}}})$, is given by

$$d_{e_1}(X, \mathcal{H}(\widehat{\theta}^{\epsilon_{\text{tr}}})) = \left| X_{[1]} + \frac{1}{\widehat{\theta}^{\epsilon_{\text{tr}}}_{[0]}} \sum_{i=2}^{d} \widehat{\theta}^{\epsilon_{\text{tr}}}_{[i]} X_{[i]} \right|.$$

Substituting the expression of $\widehat{\theta}^{\epsilon_{\text{tr}}}$ in Equation 32 yields

$$d_{e_1}(X, \mathcal{H}(\widehat{\theta}^{\epsilon_{\text{tr}}})) = \left| X_{[1]} + 2\tilde{\gamma} \frac{1}{(r - \epsilon_{\text{tr}})} \sum_{i=2}^{d} \tilde{\theta}_{[i]} X_{[i]} \right|.$$

Let $N$ be a standard normal distributed random variable. By definition $\|\tilde{\theta}\|_2^2 = 1$ and using that a sum of Gaussian random variables is again a Gaussian random variable, we can write

$$d_{e_1}(X, \mathcal{H}(\widehat{\theta}^{\epsilon_{\text{tr}}})) = \left| X_{[1]} + 2\tilde{\gamma} \frac{\sigma}{(r - \epsilon_{\text{tr}})} N \right|.$$

The robustness of $\widehat{\theta}^{\epsilon_{\text{tr}}}$ is given by the probability that $d_{e_1}(X, \mathcal{H}(\widehat{\theta}^{\epsilon_{\text{tr}}})) > \epsilon_{\text{te}}$. Hence, using that $X_1 = \pm \frac{r}{2}$ with probability $\frac{1}{2}$, we get

$$\text{Rob}(\widehat{\theta}^{\epsilon_{\text{tr}}}; \epsilon_{\text{te}}) = P\left[ \frac{r}{2} + 2\tilde{\gamma} \frac{\sigma}{(r - 2\epsilon_{\text{tr}})} N > \epsilon_{\text{te}} \right] + P\left[ \frac{r}{2} + 2\tilde{\gamma} \frac{\sigma}{(r - \epsilon_{\text{tr}})} N < -\epsilon_{\text{te}} \right]. \tag{33}$$

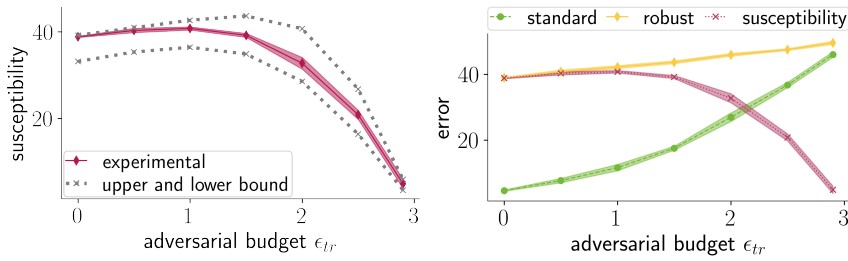

(a) Susceptibility score decreases with $\epsilon_{\text{tr}}$      (b) Robust error decomposition

Figure 7: We set $r = 6$, $d = 1000$, $n = 50$ and $\epsilon_{\text{te}} = 2.5$. (a) The average susceptibility score and the standard deviation over 5 independent experiments. Note how the bounds closely predict the susceptibility score. (b) For comparison, we also plot the robust error decomposition in susceptibility and standard error. Even though the susceptibility decreases, the robust error increases with increasing adversarial budget $\epsilon_{\text{tr}}$.

We can rewrite Equation 33 in the form

$$\text{Rob}(\widehat{\theta}^{\epsilon_{\text{tr}}}; \epsilon_{\text{te}}) = P\left[N > \frac{(r - 2\epsilon_{\text{tr}})(\epsilon_{\text{te}} - \frac{r}{2})}{2\tilde{\gamma}\sigma}\right] + P\left[N < \frac{(r - 2\epsilon_{\text{tr}})(-\epsilon_{\text{te}} - \frac{r}{2})}{2\tilde{\gamma}\sigma}\right].$$

Recall, that $N$ is a standard normal distributed random variable and denote by $\Phi$ the cumulative standard normal density. By definition of the cumulative denisity function, we find that

$$\text{Rob}(\widehat{\theta}^{\epsilon_{\text{tr}}}; \epsilon_{\text{te}}) = 1 - \Phi\left(\frac{(r - 2\epsilon_{\text{tr}})(\epsilon_{\text{te}} - \frac{r}{2})}{2\tilde{\gamma}\sigma}\right) + \Phi\left(\frac{(r - 2\epsilon_{\text{tr}})(-\epsilon_{\text{te}} - \frac{r}{2})}{2\tilde{\gamma}\sigma}\right).$$

Substituting the bounds on $\tilde{\gamma}$ of Lemma A.2 gives us the non-asymptotic bounds on the robustness score and by Equation 31 also on the susceptibility score.

## C  EXPERIMENTAL DETAILS ON THE LINEAR MODEL

In this section, we provide detailed experimental details to Figures 3 and 4.

We implement adversarial logistic regression using stochastic gradient descent with a learning rate of $0.01$. Note that logistic regression converges logarithmically to the robust max $l_2$-margin solution. As a consequence of the slow convergence, we train for up to $10^7$ epochs. Both during training and test time we solve $\max_{x_i' \in T(x_i; \epsilon_{\text{tr}})} L(f_\theta(x_i')y_i)$ exactly. Hence, we exactly measure the robust error. Unless specified otherwise, we set $\sigma = 1$, $r = 12$ and $\epsilon_{\text{te}} = 4$.

**Experimental details on Figure 3**  (a) We draw 5 datasets with $n = 50$ samples and input dimension $d = 1000$ from the distribution $\mathbb{P}$. We then run adversarial logistic regression on all 5 datasets with adversarial training budgets, $\epsilon_{\text{tr}} = 1$ to 5. To compute the resulting robust error gap of all the obtained classifiers, we use a test set of size $10^6$. Lastly, we compute the lower bound given in part 2. of Theorem 3.1. (b) We draw 5 datasets with different sizes $n$ between 50 and $10^4$. We take an input dimension of $d = 10^4$ and plot the mean and standard deviation of the robust error after adversarial and standard logistic regression over the 5 samples.(c) We again draw 5 datasets for each $d/n$ constellation and compute the robust error gap for each dataset.

**Experimental details on Figure 4**  For both (a) and (b) we set $d = 1000$, $\epsilon_{\text{te}} = 4$, and vary the adversarial training budget ($\epsilon_{\text{tr}}$) from 1 to 5. For every constellation of $n$ and $\epsilon_{\text{tr}}$, we draw 10 datasets and show the average and standard deviation of the resulting robust errors. In (b), we set $n = 50$.

## D  EXPERIMENTAL DETAILS ON THE WATERBIRDS DATASET

In this section, we discuss the experimental details and construction of the Waterbirds dataset in more detail. We also provide ablation studies of attack parameters such as the size of the motion blur kernel, plots of the robust error decomposition with increasing $n$, and some experiments using early stopping.

### D.1  THE WATERBIRDS DATASET

To build the Waterbirds dataset, we use the CUB-200 dataset Welinder et al. (2010), which contains images and labels of 200 bird species, and 4 background classes (forest, jungle/bamboo, water ocean, water lake natural) of the Places dataset Zhou et al. (2017).The aim is to recognize whether or not the bird, in a given image, is a waterbird (e.g. an albatros) or a landbird (e.g. a woodpecker). To create the dataset, we randomly sample equally many water- as landbirds from the CUB-200 dataset. Thereafter, we sample for each bird image a random background image. Then, we use the segmentation provided in the CUB-200 dataset to segment the birds from their original images and paste them onto the randomly sampled backgrounds. The resulting images have a size of $256 \times 256$. Moreover, we also resize the segmentations such that we have the correct segmentation profiles of the birds in the new dataset as well. For the concrete implementation, we use the code provided by Sagawa et al. (2020).

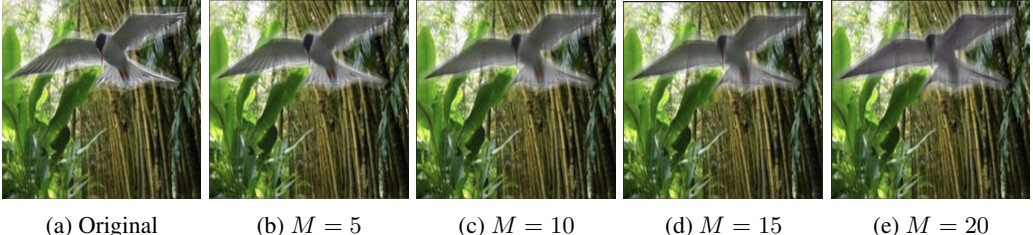

| (a) Original | (b) $M = 5$ | (c) $M = 10$ | (d) $M = 15$ | (e) $M = 20$ |

Figure 8: An ablation study of the motion blur kernel size, which corresponds to the severity level of the blur. For increasing $M$, the severity of the motion blur increases. In particular, note that for $M = 15$ and even $M = 20$, the bird remains recognizable: we do not semantically change the class, i.e. the perturbations are consistent.

## D.2    EXPERIMETAL TRAINING DETAILS

Following the example of Sagawa et al. (2020), we use a ResNet50 or ResNet18 pretrained on the ImageNet dataset for all experiments in the main text, a weight-decay of $10^{-4}$, and train for 300 epochs using the Adam optimizer. Extensive fine-tuning of the learning rate resulted in an optimal learning rate of 0.006 for all experiments using the adversarial illumination attack and a pretrained ResNet50. For the experiments considering the adversarial illumnination attack using a pretrained VGG19 or Densenet121 network, we found optimal learning rates of 0.001 and 0.002 respectively. Lastly, we found that for all experiments using the motion blur attack a learning rate of 0.0011 was optimal. Adversarial training is implemented as suggested in Madry et al. (2018): at each iteration we find the worst case perturbation with an exact or approximate method. In all our experiments, the resulting classifier interpolates the training set. We plot the mean over all runs and the standard deviation of the mean.

## D.3    SPECIFICS TO THE MOTION BLUR ATTACK

Fast moving objects or animals are hard to photograph due to motion blur. Hence, when trying to classify or detect moving objects from images, it is imperative that the classifier is robust against reasonable levels of motion blur. We implement the attack as follows. First, we segment the bird from the original image, then use a blur filter and lastly, we paste the blurred bird back onto the background. We are able to apply more severe blur, by enlarging the kernel of the filter. See Figure 8 for an ablation study of the kernel size.

The motion blur filter is implemented as follows. We use a kernel of size $M \times M$ and build the filter as follows: we fill the row $(M - 1)/2$ of the kernel with the value $1/M$. Thereafter, we use the 2D convolution implementation of OpenCV (filter2D) Bradski (2000) to convolve the kernel with the image. Note that applying a rotation before the convolution to the kernel, changes the direction of the resulting motion blur. Lastly, we find the most detrimental level of motion blur using a list-search over all levels up to $M_{max}$.

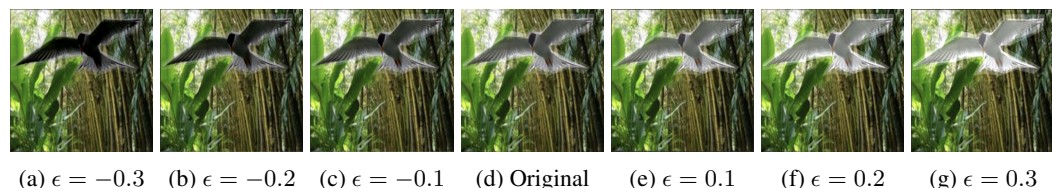

| (a) $\epsilon = -0.3$ | (b) $\epsilon = -0.2$ | (c) $\epsilon = -0.1$ | (d) Original | (e) $\epsilon = 0.1$ | (f) $\epsilon = 0.2$ | (g) $\epsilon = 0.3$ |

Figure 9: An ablation study of the different lighting changes of the adversarial illumination attack. Even though the directed attack perturbs the signal component in the image, the bird remains recognizable in all cases.

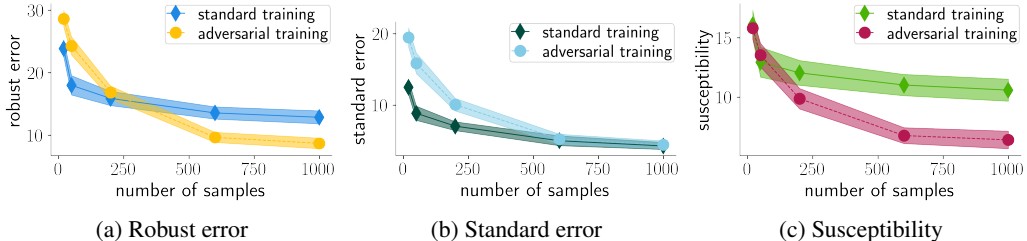

(a) Robust error        (b) Standard error        (c) Susceptibility

Figure 10: The robust error decomposition of the experiments depicted in Figure 10a. The plots depict the mean and standard deviation of the mean over several independent experiments. We see that, in comparison to standard training, the reduction in susceptibility for adversarial training is minimal in the low sample size regime. Moreover, the increase in standard error of adversarial training is quite severe, leading to an overall increase in robust error in the low sample size regime.

### D.4 Specifics to the adversarial illumination attack

An adversary can hide objects using poor lightning conditions, which can for example arise from shadows or bright spots. To model poor lighting conditions on the object only (or targeted to the object), we use the adversarial illumination attack. The attack is constructed as follows: First, we segment the bird from their background. Then we apply an additive constant $\epsilon$ to the bird, where the absolute size of the constant satisfies $|\epsilon| < \epsilon_{\text{te}} = 0.3$. Thereafter, we clip the values of the bird images to $[0, 1]$, and lastly, we paste the bird back onto the background. See Figure 9 for an ablation of the parameter $\epsilon$ of the attack. It is non-trivial how to (approximately) find the worst perturbation. We find an approximate solution by searching over all perturbations with increments of size $\epsilon_{\text{te}}/K_{\text{max}}$. Denote by seg, the segmentation profile of the image $x$. We consider all perturbed images in the form of

$$x_{pert} = (1 - seg)x + seg(x + \epsilon \frac{K}{K_{\max}} 1_{255 \times 255}), \;\; K \in [-K_{max}, K_{max}].$$

During training time we set $K_{max} = 16$ and therefore search over 33 possible images. During test time we search over 65 images ($K_{max} = 32$).

### D.5 Early stopping

In all our experiments on the Waterbirds dataset, a parameter search lead to an optimal weight-decay and learning rate of $10^{-4}$ and $0.006$ respectively. Another common regularization technique is early stopping, where one stops training on the epoch where the classifier achieves minimal robust error on a hold-out dataset. To understand if early stopping can mitigate the effect of adversarial training aggregating robust generalization in comparison to standard training, we perform the following experiment. On the Waterbirds dataset of size $n = 20$ and considering the adversarial illumination attack, we compare standard training with early stopping and adversarial training ($\epsilon_{\text{tr}} = \epsilon_{\text{te}} = 0.3$) with early stopping. Considering several independent experiments, early stopped adversarial training has an average robust error of 33.5 a early stopped standard training 29.1. Hence, early stopping does decrease the robust error gap, but does not close it.

### D.6 Error decomposition with increasing $n$

In Figure 10a and 11a, we see that adversarial training hurts robust generalization in the small sample size regime. For completeness, we plot the robust error composition for adversarial and standard training in Figure 10. We see that in the low sample size regime, the drop in susceptibility that adversarial training achieves in comparison to standard training, is much lower than the increase in standard error. Conversely, in the high sample regime, the drop of susceptibility from adversarial training over standard training is much bigger than the increase in standard error.

### D.7 Different architectures

For completeness, we also performed similar experiments on the waterbirds dataset using the adversarial illumination attack with different network architectures as with the pretrained ResNet50

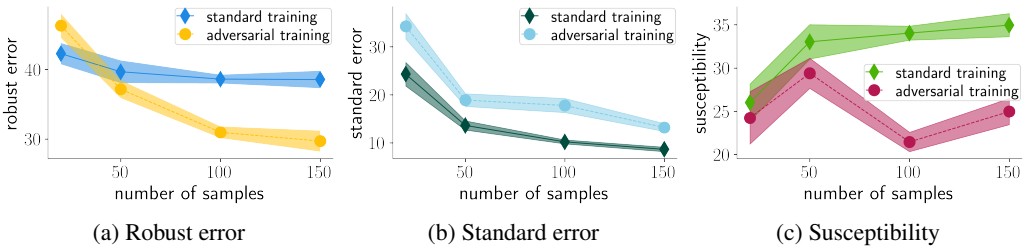

(a) Robust error      (b) Standard error      (c) Susceptibility

Figure 11: The robust error decomposition of the experiments depicted in Figure 6. The plots depict the mean and standard deviation of the mean over several independent experiments. We see that, in comparison to standard training, the reduction in susceptibility for adversarial training is minimal in the low sample size regime. Moreover, the increase in standard error of adversarial training is quite severe, leading to an overall increase in robust error in the low sample size regime.

network. In particular, we considered the following pretrained network architectures: VGG19 and Densenet121. See Figure 12 for the results. We observe that accros models, adversarial training hurts in the low sample size regime, but helps when enough data is available.

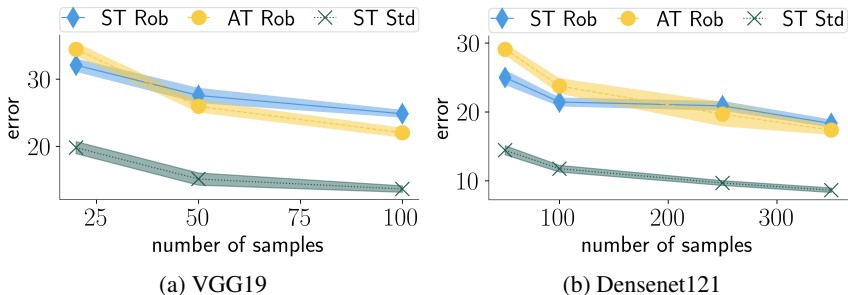

(a) VGG19      (b) Densenet121

Figure 12: The robust error of adversarial training and standard training with increasing sample size using the adversarial illumination attack with $\epsilon_{\text{te}} = 0.3$. We depict the mean and the standard deviation of the mean for multiple runs. Observe that across models, adversarial training hurts in the low sample size regime, but helps when enough samples are available.

## D.8 UNDIRECTED ATTACKS ON THE WATERBIRDS DATASET

In this section, we analyse adversarial training for $\ell_2$-and $\ell_\infty$-ball perturbations in the small sample size regime. We observe that while adversarial training hurts standard generalization, it helps robust generalization.

**Adversarial training with $\ell_2$-balls** We train and test with small $\ell_2$-balls, $\epsilon_{\text{te}} = 0.2$, such that the networks trained with standard training achieve a non-zero robust accuracy and the networks trained with adversarial training achieve non-trivial standard accuracy. We see in Figure 13, that adversarial training with $\ell_2$-balls hurts standard generalization while increasing robust generalization. Moreover, in Figure 14, we see that also in the very small sample size regime, adversarial training with increasing $\epsilon_{\text{tr}}$ increases the standard error, but reduces the susceptibility.

**Adversarial training with $\ell_\infty$-balls** We also consider $\ell_\infty$-ball perturbation. We see in Figure 15 that even the smallest perturbation budget $\epsilon_{\text{te}} = \frac{2}{255}$, standard training has robust error of 100 percent. On the other hand, adversarial training achieves low, but non-zero robust error.

**Experimental details** We use an ImageNet pretrained ResNet34 and train for 300 epochs. More-over, for reliable robust error and susceptibility evaluation of the attacks we use AutoAttack Croce & Hein (2020). All networks were trained such that the network interpolates the training dataset and has low robust error with non-trivial standard error. For the networks trained using standard training we use a learning rate of 0.006 and for the networks trained with adversarial training we used a learning

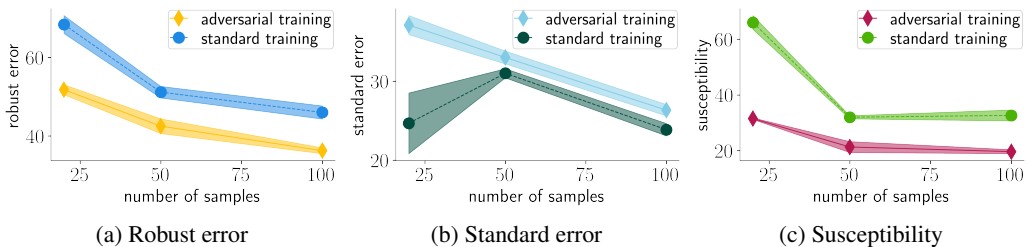

(a) Robust error  (b) Standard error  (c) Susceptibility

Figure 13: The robust error decomposition of adversarial training with $\ell_2$-balls of size $\epsilon_{\text{tr}} = 0.2$ and test adversaries with $\ell_2$-balls of size $\epsilon_{\text{te}} = 0.2$. The plots depict the mean and standard deviation of the mean over several independent experiments. We see that even though adversarial training hurts standard generalization, it increases robust generalization as it decreases the susceptibility of the classifiers.

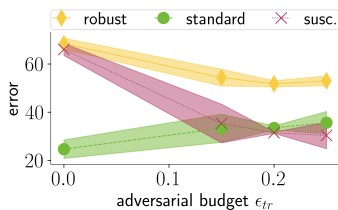

Figure 14: The robust error decomposition of adversarial training in function of $\epsilon_{\text{tr}}$ in the small sample size regime $n = 20$. We see that even though adversarial training hurts standard generalization, it increases robust generalization as it decreases the susceptibility of the classifiers with increasing $\epsilon_{\text{tr}}$. We take $n = 20$ and consider test adversaries with $\ell_2$-balls of size $\epsilon_{\text{te}} = 0.2$. The plots depict the mean and standard deviation of the mean over several independent experiments.

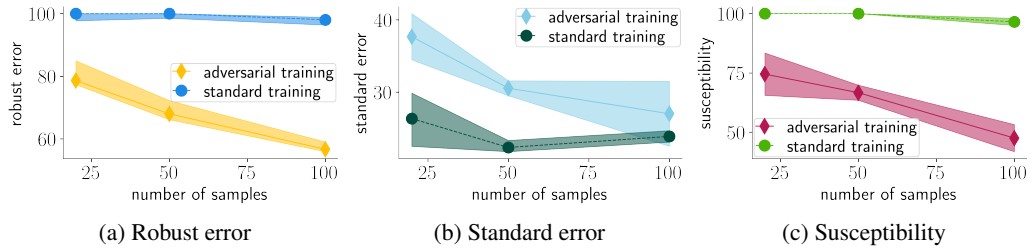

(a) Robust error  (b) Standard error  (c) Susceptibility

Figure 15: The robust error decomposition of adversarial training with $\ell_\infty$-balls of size $\epsilon_{\text{tr}} = \frac{2}{255}$ and test adversaries with $\ell_\infty$-balls of size $\epsilon_{\text{te}} = \frac{2}{255}$. The plots depict the mean and standard deviation of the mean over several independent experiments. We see that even though adversarial training hurts standard generalization, it increases robust generalization as it decreases the susceptibility of the classifiers.

rate of $5 \cdot 10^{-4}$. We also trained with a weight decay of $10^{-4}$, a batch size of 8 and a momentum of 0.9 for all networks. We train at least 3 networks for all settings and report the mean and standard deviation of the mean of the standard error, robust error and susceptibility over the three runs.

# E   EXPERIMENTAL DETAILS ON CIFAR-10

In this section, we give the experimental details on the CIFAR-10-based experiments shown in Figures 1 and 17.

**Subsampling CIFAR-10**   In all our experiments we subsample CIFAR-10 to simulate the low sample size regime. We ensure that for all subsampled versions the number of samples of each class are equal. Hence, if we subsample to 500 training images, then each class has exactly 50 images, which are drawn uniformly from the $5k$ training images of the respective class.

**Mask perturbation on CIFAR-10** On CIFAR-10, we consider the square black mask attack where the adversary can mask a square in the image of size $\epsilon_{\text{te}} \times \epsilon_{\text{te}}$ by setting the pixel values zero. To ensure that the mask cannot cover all the information about the true class in the image, we restrict the size of the masks to be at most $2 \times 2$, while allowing for all possible locations of the mask in the targeted image. For exact robust error evaluation, we perform a full grid search over all possible locations during test time. We show an example of a black-mask attack on each of the classes in CIFAR-10 in Figure 16.

During training, a full grid search is computationally intractable so that we use an approximate attack similar to Wu et al. (2020) during training time: by identifying the $K = 16$ most promising mask locations with a heuristic as follows. First, we identify promising mask locations by analyzing the gradient, $\nabla_x L(f_\theta(x), y)$, of the cross-entropy loss with respect to the input. Masks that cover part of the image where the gradient is large, are more likely to increase the loss. Hence, we compute the $K$ mask locations $(i, j)$, where $\|\nabla_x L(f_\theta(x), y)_{[i:i+2, j:j+2]}\|_1$ is the largest and take using a full list-search the mask that incurs the highest loss. Our intuition from the theory predicts that higher $K$, and hence a more exact "defense", only increases the robust error of adversarial training, since the mask could then more efficiently cover important information about the class.

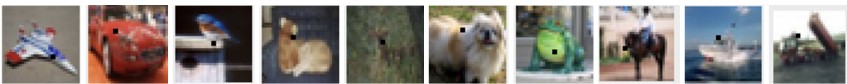

Figure 16: We show an example of a mask perturbation for all 10 classes of CIFAR-10. Even though the attack occludes part of the images, a human can still easily classify all images correctly.

**Experimental training details** For all our experiments on CIFAR-10, we adjusted the code provided by Phan (2021). As typically done for CIFAR-10, we augment the data with random cropping and horizontal flipping. For the experiments with results depicted in Figures 1 and 17, we use a ResNet18 network and train for 100 epochs. We tune the parameters learning rate and weight decay for low robust error. For standard standard training, we use a learning rate of $0.01$ with equal weight decay. For adversarial training, we use a learning rate of $0.015$ and a weight decay of $10^{-4}$. We run each experiment three times for every dataset with different initialization seeds, and plot the average and standard deviation over the runs.

For the experiments in Figure 1 and 18 we use an attack strength of $K = 4$. Recall that we perform a full grid search at test time and hence have a good approximation of the robust accuracy and susceptibility score.

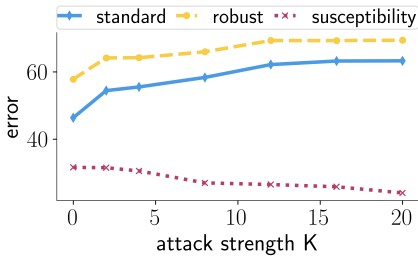

Figure 17: The robust error decomposition in standard error and susceptibility for varying attack strengths $K$. We see that the larger $K$, the lower the susceptibility, but the higher the standard error.

**Increasing training attack strength** We investigate the influence of the attack strength $K$ on the robust error for adversarial training. We take $\epsilon_{\text{tr}} = 2$ and $n = 500$ and vary $K$. The results are depicted in Figure 17. We see that for increasing $K$, the susceptibility decreases, but the standard error increases more severely, resulting in an increasing robust error.

**Robust error decomposition** In Figure 1, we see that the robust error increases for adversarial training compared to standard training in the low sample size regime, but the opposite holds when enough samples are available. For completeness, we provide a full decomposition of the robust error in standard error and susceptibility for standard and adversarial training. We plot the decomposition in Figure 18.

## F    STATIC HAND GESTURE RECOGNITION

The goal of static hand gesture or posture recognition is to recognize hand gestures such as a pointing index finger or the okay-sign based on static data such as images Oudah et al. (2020); Yang et al.

(2013). The current use of hand gesture recognition is primarily in the interaction between computers and humans Oudah et al. (2020). More specifically, typical practical applications can be found in the environment of games, assisted living, and virtual reality Mujahid et al. (2021). In the following, we conduct experiments on a hand gesture recognition dataset constructed by Mantecón et al. (2019), which consists of near-infrared stereo images obtained using the Leap Motion device. First, we crop or segment the images after which we use logistic regression for classification. We see that adversarial logistic regression deteriorates robust generalization with increasing $\epsilon_{\text{tr}}$.

**Static hand-gesture dataset**   We use the dataset made available by Mantecón et al. (2019). This dataset consists of near-infrared stereo images taken with the Leap Motion device and provides detailed skeleton data. We base our analysis on the images only. The size of the images is $640 \times 240$ pixels. The dataset consists of 16 classes of hand poses taken by 25 different people. We note that the variety between the different people is relatively wide; there are men and women with different posture and hand sizes. However, the different samples taken by the same person are alike.

We consider binary classification between the index-pose and L-pose, and take as a training set 30 images of the users 16 to 25. This results in a training dataset of 300 samples. We show two examples of the training dataset in Figure 19, each corresponding to a different class. Observe that the near-infrared images darken the background and successfully highlight the hand-pose. As a test dataset, we take 10 images of each of the two classes from the users 1 to 10 resulting in a test dataset of size 200.

**Cropping the dataset**   To speed up training and ease the classification problem, we crop the images from a size of $640 \times 240$ to a size of $200 \times 200$. We crop the images using a basic image segmentation technique to stay as close as possible to real-world applications. The aim is to crop the images such that the hand gesture is centered within the cropped image.

For every user in the training set, we crop an image of the L-pose and the index pose by hand. We call these images the training masks $\{\text{masks}_i\}_{i=1}^{20}$. We note that the more a particular window of an image resembles a mask, the more likely that the window captures the hand gesture correctly. Moreover, the near-infrared images are such that the hands of a person are brighter than the surroundings of the person itself. Based on these two observations, we define the best segment or window, defined by the upper left coordinates $(i, j)$, for an image $x$ as the solution to the following optimization problem:

$$\underset{i \in [440], \ j \in [40]}{\arg \min} \sum_{l=1}^{20} \|\text{masks}_l - x_{\{i:i+200,j:j+200\}}\|_2^2 - \frac{1}{2} \|x_{\{i+w,j+h\}}\|_1. \tag{34}$$

Equation 34 is solved using a full grid search. We use the result to crop both training and test images. Upon manual inspection of the cropped images, close to all images were perfectly cropped. We replace the handful poorly cropped training images with hand-cropped counterparts.

**Square-mask perturbations**   Since we use logistic regression, we perform a full grid search to find the best adversarial perturbation at training and test time. For completeness, the upper left coordinates

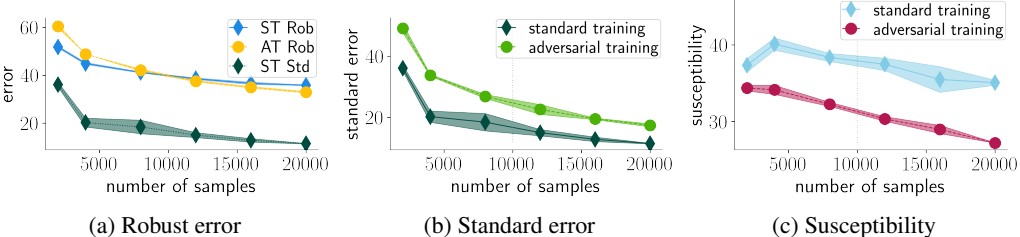

(a) Robust error          (b) Standard error          (c) Susceptibility

Figure 18: The robust error decomposition in standard error and susceptibility of the subsampled datasets of CIFAR-10 after adversarial and standard training. For small sample size, adversarial training has higher robust error then standard training.

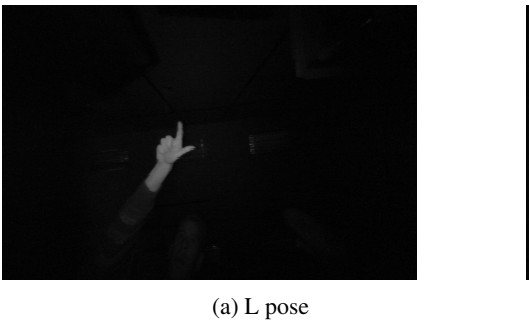
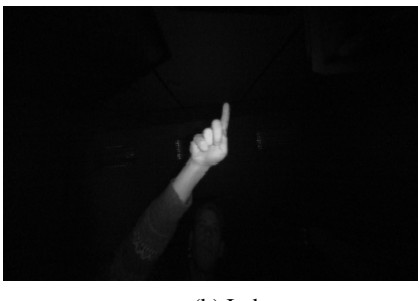

(a) L pose                              (b) Index pose

Figure 19: Examples of the original images of the considered hand-gestures. We recognize the "L"-sign in Figure 19a and the index sign in Figure 19b. Observe that the near-infrared images highlight the hand pose well and blends out much of the non-useful or noisy background.

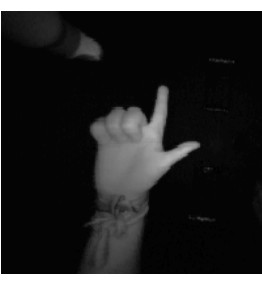
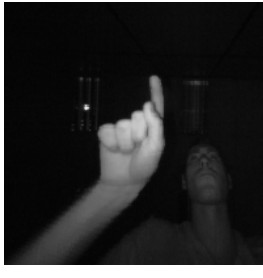
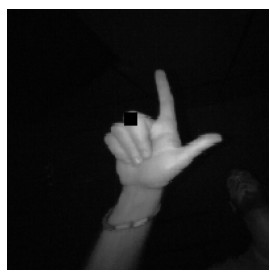

(a) Cropped L pose            (b) Cropped index pose        (c) Black-mask perturbation

Figure 20: Examples of the cropped hand-gesture images. We see that the hands are centered and the images have a size of $200 \times 200$. In Figure 20c we show an example of the square black-mask perturbation.

of the optimal black-mask perturbation of size $\epsilon_{\text{tr}} \times \epsilon_{\text{tr}}$ can be found as the solution to

$$\arg\max_{i \in [200 - \epsilon_{\text{tr}}], \ j \in [200 - \epsilon_{\text{tr}}]} \sum_{l,m \in [\epsilon_{\text{tr}}]} \theta_{[i:i+l, j:j+m]}. \tag{35}$$

The algorithm is rather slow as we iterate over all possible windows. We show a black-mask perturbation on an $L$-pose image in Figure 20c.

**Results** We run adversarial logistic regression with square-mask perturbations on the cropped dataset and vary the adversarial training budget and plot the result in Figure 21. We observe attack that adversarial logistic regression deteriorates robust generalization.

Because we use adversarial logistic regression, we are able to visualize the classifier. Given the classifier induced by $\theta$, we can visualize how it classifies the images by plotting $\frac{\theta - \min_{i \in [d]} \theta_{[i]}}{\max_{i \in [d]} \theta_{[i]}} \in [0,1]^d$. Recall that the class-prediction of our predictor for a data point $(x, y)$ is given by $\text{sign}(\theta^\top x) \in \{\pm 1\}$. The lighter parts of the resulting image correspond to the class with label 1 and the darker patches with the class corresponding to label $-1$.

We plot the classifiers obtained by standard logistic regression and adversarial logistic regression with training adversarial budgets $\epsilon_{\text{tr}}$ of 10 and 25 in Figure 22. The darker parts in the classifier correspond to patches that are typically bright for the $L$-pose. Complementary, the lighter patches in the classifier correspond to patches that are typically bright for the index pose. We see that in the case of adversarial logistic regression, the background noise is much higher than for standard logistic regression. In other words, adversarial logistic regression puts more weight on non-signal parts in the images to classify the

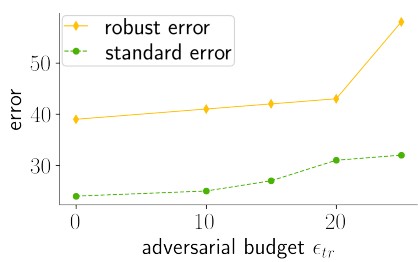

Figure 21: The standard error and robust error for varying adversarial training budget $\epsilon_{\text{tr}}$. We see that the larger $\epsilon_{\text{tr}}$ the higher the robust error.

training dataset and hence exhibits worse performance on the test dataset.

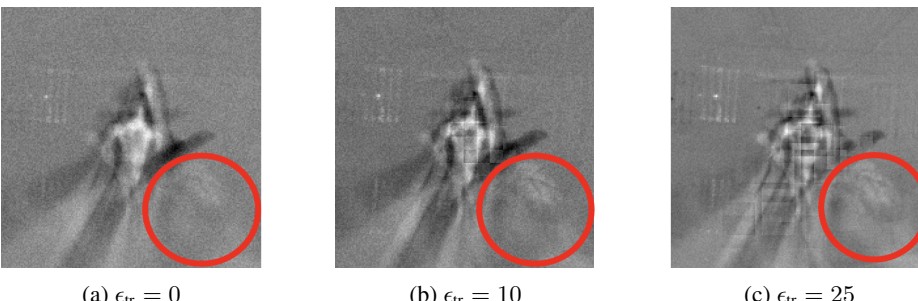



(a) $\epsilon_{\mathrm{tr}} = 0$        (b) $\epsilon_{\mathrm{tr}} = 10$        (c) $\epsilon_{\mathrm{tr}} = 25$



Figure 22: We visualize the logistic regression solutions. In Figure 22a we plot the vector that induces the classifier obtained after standard training. In Figure 22b and Figure 22c we plot the vector obtained after training with square-mask perturbations of size 10 and 25, respectively. We note the non-signal enhanced background correlations at the parts highlighted with the red circles in the image projection of the adversarially trained classifiers.

