# OpenReview forum: "Why adversarial training can hurt robust accuracy"
_ICLR.cc/2023/Conference — ICLR 2023 poster_

### Official Review · Reviewer_udiw · 2022-10-24

**Confidence:** 4
**Correctness:** 4
**Technical Novelty And Significance:** 4
**Empirical Novelty And Significance:** 4
**Recommendation:** 8

**Clarity, Quality, Novelty And Reproducibility:**

- Clearly written and presented.
- Novel behavior and analysis.

**Strength And Weaknesses:**

This is an interesting paper, which I enjoyed reading very much. The authors clearly study and characterize the simple linear setting, and their conclusions seem to extend, empirically, to more complicated scenarios.

I have a few comments, mostly minor:

1. The definition and analysis of "directed attack" is interesting, as it is a relative new concept (to my knowledge). I understand why restricting their analysis to these cases in particular is important. Yet, could the authors comment more on the differences between regular/traditional adversarial losses (say, L2 or Linfty bounded) and their "directed attack" model? It seems to me that in the linear case with only 1 informative feature, they would be equivalent (except in the Linfty case), but maybe I'm missing something.

2. In their definition of logistic loss, immediately after Eq 2, i think the authors meant $e$ instead of $\mathbb E$ (expectation operator).

3. Commenting on Eq (7), the authors mention that this equation suggests that the robust error "can only be small if both the standard error and susceptibility are small". This is not true, as Eq. (7) is just an upper bound on the robust error. Probably the authors meant "a small standard error and susceptibility imply a small robust error". At the same time, it seems like that for the linear model studied here this bound could be tight?

**Summary Of The Paper:**

This paper studies the peculiar observation that, with very sample sizes, robust training can be detrimental for robust error. The authors show that in a classification task given by a linear model, the robust risk of the max-margin classier can increase as a function of the perturbation size used during robust training. Importantly, they study a new notion of robust training that they call "directed attacks", which specifically targets the informative features of the problem. The authors then show how, in more complex settings where analysis is harder (and not presented here) some of the intuitions gained for the linear case seem to translate as well. The paper is clearly written and organized.


**Summary Of The Review:**

This paper studies a new (to my knowledge) behavior in adversarial training and shows interesting results.

---

> ### Author Response · Authors · 2022-11-10
> **Response to reviewer**
>
> Dear Reviewer,
>
> Thank you for appreciating our contributions. Regarding the following three comments.
>
> >The definition and analysis of "directed attack" is interesting, as it is a relatively new concept (to my knowledge). I understand why restricting their analysis to these	cases in particular is important. Yet, could the authors comment more on the differences between regular/traditional adversarial losses (say, $\ell_2$ or $\ell_\infty$ bounded) and their "directed attack" model? It seems to me that in the linear case with only 1 informative feature, they would be equivalent (except in the $\ell_\infty$ case), but maybe I'm missing something.
>
> Thanks a lot for your question. We assume that you meant to write “traditional adversarial perturbations” In the linear model, $\ell_1$ perturbations would indeed yield the same perturbation direction (Corollary 3.2.) For $\ell_2$ perturbations, it is known that adversarial logistic regression with $\ell_2$ perturbations yields the identical classifier as standard logistic regression on robustly separable data (see Liu 2020). Therefore there is no robust performance gap for $\ell_2$. For $\ell_p$ perturbations with $p>2$, we can prove that in our setting adversarial logistic regression yields models with a lower robust error than standard logistic regression. If it is helpful, we can add a proof in the final revision. For images, we explicitly discuss the relation to $\ell_p$ bounded attacks in the introduction: $\ell_p$ perturbations do not necessarily focus all their attention on the object, i.e. they are more “targeted” towards the distinctive features than a general $\ell_p$ ball perturbation. In the introduction we write “They have the distinguishing property to effectively reduce actual class information in the input without necessarily changing the true label”. In our Waterbirds experiments in Appendix D.8. We indeed perform $\ell_p$ perturbation experiments and show that the phenomenon does not occur for such “undirected” attacks.
>
> >In their definition of logistic loss, immediately after Eq 2, i think the authors meant e instead of E (expectation operator).
>
> Yes, thank you for pointing this typo out. We revised it.
>
> >Commenting on Eq (7), the authors mention that this equation suggests that the robust error "can only be small if both the standard error and susceptibility are small". This is not true, as Eq. (7) is just an upper bound on the robust error. Probably the authors meant "a small standard error and susceptibility imply a small robust error". At the same time, it seems like that for the linear model studied here this bound could be tight?
>
> Thanks a lot for the question. We decided to use the word  “suggest” to indicate that it is, in general, not a “tight” statement (i.e. missing lower bound). At the same time, you are correct that for our noiseless linear setting, this equation is almost tight with a lower bound given by $Err(\theta; 0) + 0.5 Susc(\theta; \epsilon_{te})$. As we wanted to write the proof intuition such that it can be transferred to more general settings (including the image classification one), we opted for a less rigorous statement here.  However, we understand that this is misleading as it stands and hence added the lower bound for the linear setting.
>
> Kind regards,
> the authors

---

> > ### Comment · Reviewer_udiw · 2022-11-28
> > **Thank you for the clarifications**
> >
> > Dear authors - thank you for your clarifying responses, which have fully answered my questions.

---

### Official Review · Reviewer_7vDF · 2022-10-24

**Confidence:** 5
**Correctness:** 3
**Technical Novelty And Significance:** 4
**Empirical Novelty And Significance:** 4
**Recommendation:** 6

**Clarity, Quality, Novelty And Reproducibility:**

The quality and novelty are both good subject to straightening up the presentation. Please see below for some comments and suggestions.

- Title
  - Does the scope of this study, e.g. as outlined in Section 1, actually answer the general question posed in the title?
    - The paper concerns a "new" special class of adversarial attacks, and its findings specifically hold in the low-data regime.
    - As such, it is not clear how the findings reported relate to the mainstream work on adversarial robustness in the absence of those essential conditions.
- Abstract
  - I strongly recommend to limit the use of sentiment words such as "surprising". It appears twice in the abstract.
    - First occurrence is justified with respect to the common belief.
    - Second occurrence is unjustified: it is the opposite of surprising to apply insights from a mathematical proof.
  - Please use the terms perturbations or attacks consistently, especially in the abstract.
- Section 1
  - Could you please motivate the new term given to "directed" attacks?
    - What are they directed to?
    - How does this concept relate to the more commonly used term of "targeted" attacks, where a given misclassification is desired?
    - Perhaps something like "hindering" attacks could be more accurate.
      - Indeed, "corruption attacks" appears early in Section 4.
  - Figure 1: reference to a green curve?
  - Please use the terms robust error, or adversarial error, or robust test error consistently, especially in the introduction.
  - I strongly recommend to rewrite the penultimate paragraph to read more easily. I didn't find the visual effects to be helpful to my understanding, especially as the sentence structure with added citations is a bit complicated already.
    - If absolutely necessary, then "in the low-sample regime" should be included in the punch line.
  - The 2nd point in the contribution list seems like merely a bridge from the 1st to the 3rd, and if so, should probably be removed. If there's more technical novelty to it, e.g., how linear separability maps to the low-data regime, then please clarify. Intuition can make for valuable discussion, but does not stand on its own as a contribution.
- Section 2
  - Equation 1
    - Please define $\mathbb{P}$
    - There is no indication of the transformation type in the definition of $T(x; \epsilon_{te})$
      - This is more of an issue when this notation is used later to indicate the same perturbation set for training and testing writing $T(x; \epsilon_{tr}) = T(x; \epsilon_{te})$
    - The definition given for $\ell$ only applies for class probabilities, and not the sign interpretation for the binary case.
  - Directed attacks
    - attack of the (input $x$)? .. for the model $f_\theta$
    - Is the direction of the optimal decision boundary the reason of calling those directed attacks? If so, how does that correspond to the characterization that such attacks "effectively reduce the information about the true class"? or to the image manipulations as shown in Figure 2?
  - It was probably meant to write $L(z) = \log(1 + e^{-z})$ not $\mathbb{E}^{-z}$.
- Section 3
  - What is the "true signal"?
  - Data model
    - $d - 1 > n$, as in Theorem 3.1, makes $n$ lower than the ambient dimension d minus 1.
    - Please clarify that $e_1$ is the first standard coordinate vector.
    - Please clarify what $u_1$ is.
    - Please clarify the last sentence, seeing that consistent perturbations are not defined and the role of $r$ has not been explained.
      - $r$ is only referred to as the "separation" early in $3.2.
  - Robust max-$\ell_2$-margin classifier
    - What is the relevance of the implicit bias of interpolators to the development?
    - Please motivate the choice of the robust max-margin solution as written in Eq. 4.
  - Main results
    - The statement of Theorem 3.1 needs to be rephrased. It was going in the direction of saying "the $\epsilon_{te}$-robust error ..?" but ended up saying "the following holds"
    - It appears you can safely move the definitions of $\phi_{min}$ and $\phi_{max}$ to the statement of the theorem.
    - Equation 5 is not really usable as an equation with $\tilde{\phi}$ unspecified. Part (a) as presented only serves the body of the paper to establish a monotonic relationship with the robust training budget $\epsilon_{tr}$, and as such, I recommend to present it as such.
    - ~~Please indicate in the statement of the theorem for part (b) that it assumes a fixed $\epsilon_{tr} = \epsilon_{te}$.~~
    - Why the large values of $\epsilon \in [0, 5]$ for the experiments in this section? It seems that for the experiments on images $\epsilon \in [0, 0.5]$.
  - Proof intuition
    - the solution of adversarial training => $\epsilon_{te}$-robust margin ~ Eq. (1)
    - $\epsilon_{tr}$ appears nowhere near Equation 7, yet $\text{Susc}(\theta; \epsilon_{te})$ is defined as the $\epsilon_{tr}$-attack-susceptibility.
      - I recommend you recall Equation 2 explaining how $\epsilon_{tr}$ is implicit in the training objective to obtain the $\theta$ in Equation 7.
    - ~~$\hat{\theta}^{\epsilon_{tr}}$ is used before it was defined.~~
- Nitpicking
  - Please use the technical term consistently
    - "small sample size" (abstract), "low-sample regime" (Section 1, Section 4), "low sample size" (Section 4), "low-sample size" (Section 6).
    - normal vs. Gaussian
  - Section 1
    - Please use the correct citation type, and fix up the punctuation
      - human faces (Wu et al. 2020))
      - translations or corruptions Engstrom et al. (2019)
      - as noted in Zhang et al. (2019) .... ")"
  - Section 3
    - in function of => as a function of (both in the text and the caption of Figure 3)
  - Section 4
    - appply

**Strength And Weaknesses:**

- Strengths
  - Brings attention to a theoretically-interesting aspect of adversarial training, that could be relevant in practice.
  - Presents theoretical analysis in the linear 2-class case, verified by corresponding experiments.
  - Applies the results to a restricted class of adversarial attacks on image datasets, (claiming to) corroborate the theory.
- Weaknesses
  - The presentation needs significant work ~~,to the point that I'm unable to grasp the core technical points or the empirical evaluations.~~
    - The definition of directed attacks, involving notions of true signal and class information, needs to be solidified with a clear and consistent linking to the Bayes optimal classifier.
    - The qualification of the implemented attacks in the image context as instance of directed attacks needs to be established more rigorously.
    - Important recent work needs to be cited and discussed to better position the present contributions; see below.

**Summary Of The Paper:**

The authors study the impact of adversarial training on both the clean and robust error in the low-sample regime using a special type of adversarial attacks, termed directed attacks. The main message is that adversarial training may even hurt the robust accuracy, in addition to the well-known effect of reducing the clean accuracy. The authors analyze the linearly-separable binary classification problem, where directed attacks are easily defined in relation to the optimal separator, and use the robust max-$\ell_2$-margin suggesting to decompose the robust error into a clean error term and a susceptibility term. The analysis, along with experiments, is aimed to elucidate the effect of the train/test budgets on each type of error. The findings of this analysis are then extended to the multi-class image classification problem using deep CNNs where directed attacks are generalized to encompass a number of image corruptions. Results on standard and customized image datasets are claimed to corroborate the theoretical and empirical findings from earlier.

**Summary Of The Review:**

The authors clearly put significant work, evident by the abundance of content and auxiliary analyses. However, the presentation needs significant work to polish up the key contributions and make the results more accessible.

Following my extended discussion with the authors, and taking the time to look up more of the recent literature, I'm updating my evaluation with the following requests:
- Please revise the definition of directed attacks, involving notions of true signal and class information, with a clear and consistent linking to the Bayes optimal classifier.
- Please include a careful derivation or additional empirical evidence to the qualification of the implemented attacks in the image context as instance of directed attacks.
- Please cite and discuss related recent works in the same vein of the generalization and accuracy-robustness trade-offs of adversarial training, as I list below. It appears the present work is distinguished by more insightful empirical contributions, while similar works are either mostly theoretical or only include limited experiments.
  - I recommend to discuss the following papers, which seemed most relevant to the main questions studied:
>- Dobriban, Edgar, Hamed Hassani, David Hong, and Alexander Robey. "Provable tradeoffs in adversarially robust classification." arXiv preprint arXiv:2006.05161 (2020).
>- Min, Yifei, Lin Chen, and Amin Karbasi. "The curious case of adversarially robust models: More data can help, double descend, or hurt generalization." In Uncertainty in Artificial Intelligence, pp. 129-139. PMLR, 2021.
>- Dong, Chengyu, Liyuan Liu, and Jingbo Shang. "Data Quality Matters For Adversarial Training: An Empirical Study." arXiv preprint arXiv:2102.07437 (2021).
>- Mendonça, Marcele OK, Javier Maroto, Pascal Frossard, and Paulo SR Diniz. "Adversarial training with informed data selection." In 2022 30th European Signal Processing Conference (EUSIPCO), pp. 608-612. IEEE, 2022.
  - I recommend to include a brief citation of the following papers:
>- Attias, Idan, Aryeh Kontorovich, and Yishay Mansour. "Improved generalization bounds for robust learning." In Algorithmic Learning Theory, pp. 162-183. PMLR, 2019.
>- Xing, Yue, Qifan Song, and Guang Cheng. "On the generalization properties of adversarial training." In International Conference on Artificial Intelligence and Statistics, pp. 505-513. PMLR, 2021.

With a better understanding of the positioning of the presented contributions, I am increasing my score with residual concerns regarding the clarity of presentation and the anticipated impact given those clarity issues.

---

> ### Author Response · Authors · 2022-11-11
> **Response to reviewer abstract and title**
>
> Dear Reviewer,
>
> Thank you for the very detailed review, appreciating the contributions of the paper and the many suggestions. We will upload the final revised submission after the discussion and present the planned changes here, that address your main doubts about the representation - including a slightly different choice of words, sentence structure, sentence rephrasing and some typos. Further, we discuss your doubts about the term “directed” (and are open to alternative suggestions), the title, the relation to mainstream adversarial robustness literature and the importance of our Theorem depending on $\epsilon_{tr}$ vs. $\epsilon_{te}$ separately (not assuming that they are equal) and also answer other clarification questions in your review.
> If there are any particular aspects that are still unaddressed and that are the reason for your low score, we would be grateful if you could elaborate on them so that we can incorporate them in the revision. We now give in-line responses to your comments.
>
> >Does the scope of this study, e.g. as outlined in Section 1, actually answer the general question posed in the title? The paper concerns a "new" special class of adversarial attacks, and its findings specifically hold in the low-data regime.
>
> We believe the word “can” points towards the existence of conditions for which the phenomenon occurs. In the paper abstract, we characterize exactly which conditions these are (low sample regime and directed attacks) and give a proof as well as a proof intuition that explains the why. Hence we believe that the title is justified in its current form. If you and other reviewers perceive the words differently, we are open to other suggestions.
>
> >As such, it is not clear how the findings reported relate to the mainstream work on adversarial robustness in the absence of those essential conditions.
>
> That is exactly our point: The aim of this work is to draw attention to the fact, that some wisdoms that have manifested in the community based on the (very limited) mainstream types of adversarial robustness (such as $\ell_p$) do not generalize for some other quite natural types of attacks / robustness (mask attacks have been studied before, in the context of traffic signs, motion blurs may happen naturally in practice). We make this point in Section 4.3 - 4.4.
>
> **Abstract**
> >I strongly recommend to limit the use of sentiment words such as "surprising". It appears twice in the abstract. First occurrence is justified with respect to the common belief. Second occurrence is unjustified: it is the opposite of surprising to apply insights from a mathematical proof. Please use the terms perturbations or attacks consistently, especially in the abstract.
>
> Indeed, the second “surprisingly” is unnecessary and we deleted it. We now only use the term “perturbations” in the abstract

---

> ### Author Response · Authors · 2022-11-11
> **Response to reviewer Section 1**
>
> **Section 1**
> >Could you please motivate the new term given to "directed" attacks? What are they directed to?
>
> We define directed attacks in Section 2 generally as attacks that effectively reduce the information about the true class label in the image and give examples, i.e. they are directed against the “signal” in the image. To clarify further:
> The true class label is usually the label of some object in the image. We say that directed attacks are the ones that use a large budget to attack the class information in an input.
> Mathematically, e.g. for binary classification as in our setting in Section 3.1,  “effectively reducing the information about the class label” can be modeled as moving in the direction “orthogonal to” the optimal decision boundary. Hence directed.
> In the case of image classification as in Section 4, reducing the information of the true class in an image corresponds to directly distorting parts of the object that are most distinctive. For example in the motion blur case, the fine predictive features of the birds are blurred. Hence, even though the birds are still recognizable, the classifier cannot use these fine-grained features to classify them correctly.
>
> >How does this concept relate to the more commonly used term of "targeted" attacks, where a given misclassification is desired?
>
> Targeted attacks are different: a targeted attack is an attack that aims to find a perturbation \delta within a predefined perturbation set T(x, epstest) such that the perturbed image is classified as a certain class y different from the true class label. In our case, for one we do not define a certain target class, and second the attack is such that the distinctive features of the class are directly “diminished” (see above paragraph)
>
> >Perhaps something like "hindering" attacks could be more accurate. Indeed, "corruption attacks" appears early in Section 4.
>
> In some sense all attacks (including $\ell_2$ or $l_\infty$) are “hindering” - by directed attacks we mean the attacks that use the entire budget to move on the shortest path to the decision boundary. Having said that, we admit the term “directed” is not optimal, however, we were not able to come up with a better one. If all reviewers agree on a term that is clearer than “directed”, we would be happy to consider using it instead.
>
> >Figure 1: reference to a green curve?
>
> Apologies for this typo, we corrected it.
>
> >Please use the terms robust error, or adversarial error, or robust test error consistently, >especially in the introduction.
>
> Both terms adversarial and robust error are often used in the literature to denote the “adversarially” robust error on the test set (for images) or the robust population error in theoretical papers (as defined in Equation 1). We used the robust test error in the introduction which might have caused some confusion  -we corrected it. For simplicity, we now only use the term “robust error”.
>
> >I strongly recommend to rewrite the penultimate paragraph to read more easily. I didn't find the visual effects to be helpful to my understanding, especially as the sentence structure with added citations is a bit complicated already. If absolutely necessary, then "in the low-sample regime" should be included in the punch line.
>
> We note that the phrase “in the low sample size regime” is in the punch-line phrase already, but not part of the italic sentence for the simple fact that the sentence would not fit in one line. However, if the reviewers agree that it is helpful, we are happy to add it back. We modified the sentence structure and changed the citation style in the paragraph to increase readability.
>
> >The 2nd point in the contribution list seems like merely a bridge from the 1st to the 3rd, and if so, should probably be removed. If there's more technical novelty to it, e.g., how linear separability maps to the low-data regime, then please clarify. Intuition can make for valuable discussion, but does not stand on its own as a contribution.
>
> We understand your sentiment. Here’s our reasoning for including it as a bullet point: Theorems do not always yield an intuition that can translate beyond the restricted setting. By adding the bullet point, we wanted to stress that the main proof idea of our theorem indeed does. In fact, the theorem existed first and it was quite surprising that using its proof intuition, we were able to identify realistic attacks for images that yield the same phenomenon. However, if all reviewers agree that this is not a valid bullet point, we can consider removing it.

---

> > ### Comment · Reviewer_7vDF · 2022-11-12
> > **Thanks for updating the manuscript**
> >
> > I will re-read the (updated) manuscript and try to get to the bottom of it.

---

> > ### Comment · Reviewer_7vDF · 2022-11-12
> > **Updated manuscript?**
> >
> > You are not required to, but I thought you updated the manuscript since you talked about implementing suggested changes. If you have a revised manuscript, could you please update it?

---

> > > ### Author Response · Authors · 2022-11-16
> > > **Updated manuscript**
> > >
> > > Dear reviewer,
> > > We wanted to update the manuscript only after we’ve heard back from you about some of the questions for clarification. We have now uploaded a new version with all the changes and leave new comments.
> > > Thanks a lot again for your efforts!

---

> ### Author Response · Authors · 2022-11-11
> **Response to reviewer Section 2**
>
> **Section 2**
>
> > Please define $\mathbb{P}$
>
> Indeed, after restructuring the paper, the symbol $\mathbb{P}$ ended up appearing in Eq. 1 before it was mentioned - we now add that samples x,y at both training and test time are assumed to be draws from $\mathbb{P}$.
>
> > There is no indication of the transformation type in the definition of $T(x, \epsilon_{tr})$. This is more of an issue when this notation is used later to indicate the same perturbation set for training and testing writing $T(x, \epsilon_{tr}) = T(x, \epsilon_{te})$
>
> We use $T$ to denote general perturbation types which are instantiated for particular instances.  In the standard AT setting, the same perturbation types are used during training and test time. Sometimes the test time perturbation type is unknown and hence training uses a different one. We discuss this in Section 1, 3.3 and 3.4.
>
> >The definition given for $\ell$ only applies for class probabilities, and not the sign interpretation for the binary case.
>
> Indeed, we adjusted the definition of the 0-1 loss \ell in Equation 1 to also hold for the sign-interpretation in the binary case.
>
> >It was probably meant to write $L(z) = 1/\log e^{-z}$ not $\mathbb{E}$
>
> Thank you for pointing out the typo in the definition of L(z). We corrected it.

---

> > ### Comment · Reviewer_7vDF · 2022-11-12
> > **Notation: transformation type**
> >
> > On P.2
> > > Further, $T(x; \epsilon_{te})$ is a perturbation set defined by a *transformation type* and size $\epsilon_{te}$.
> >
> > Perhaps you can say that the transformation type is implicit and will be clear in the context; which slightly abuses the notation. Alternatively, you may consider defining a family of transformations of type $\mathcal{F}$ and write $T_\mathcal{F}(x; \epsilon_{te})$; this way you would be able to specify the transformation type $\mathcal{F}$ or change it to some other $\mathfrak{F}$.

---

> > > ### Author Response · Authors · 2022-11-16
> > > **Notation: transformation type answer**
> > >
> > > >Perhaps you can say that the transformation type is implicit and will be clear in the context; which slightly abuses the notation. Alternatively, you may consider defining a family of transformations of type $F$ and write $T_F(x;\epsilon_{te})$; this way you would be able to specify the transformation type $F$ or change it to some other $F$.
> > >
> > > Thanks for clarifying your question further. Indeed currently, the sentence that starts right after Equation 1 can be slightly misleading. It should read: “Further, $T(x;\epsilon_{te})$ indicates a perturbation set around $x$ of a certain transformation type (that we vary for different scenarios in the paper) with size $\epsilon_{te}$.” Would this be clearer to you? We believe with this modification it is not an abuse of notation. For example, it’s common to write a risk $R(\theta)$ or $R(\hat{\theta})$ in terms of arbitrary $\theta$ or $\hat{\theta}$ and instantiate it with a particular estimator $\hat{\theta}$ when they are discussed - without necessarily adding a subscript etc.

---

> ### Author Response · Authors · 2022-11-11
> **Response to reviewer Section 3 part A**
>
> **Section 3**
>
> >What is the "true signal"?
>
> The true signal is the signal component in the covariate vector x. In our case, this is the first coordinate of the covariates. We changed to using "the signal component".
>
> >$d-1 > n$ , as in Theorem 3.1, makes $n$  lower than the ambient dimension d minus 1
>
> Indeed, d-1 > n implies that n is smaller than the dimension “minus one”, and hence smaller than the dimension. Hence our sentence was correct.  For clarity, we added “minus one” in this sentence.
>
> >Please clarify that $e_1$ is the first standard coordinate vector. Please clarify what $u_1$ is.
>
> We use standard notation and write $e_1$ for the first standard canononical basis vector $(1,0,...,0)$ and $u_1$ is the first column vector of the matrix $U$. We now added the definitions for completeness, thanks for pointing out that they were missing.
>
> >Please clarify the last sentence, seeing that consistent perturbations are not defined and the role of $r$ has not been explained.
> >$r$ iss only referred to as the "separation" early in $3.2.
>
> Thanks for spotting the suboptimally formulated sentence. We will change it to: “By definition, the ground truth is robust against all consistent perturbations and hence the optimal robust classifier”. We define consistent perturbations in the second sentence of the introduction. However, we agree that it would be worth defining it more formally in this section as perturbations that do not change the ground truth. We will include that in the revision of Section 3.  It is also a good idea to add in words (which we do in the revision) what the role of r is - it models the strength of the signal in the data.
>
> >What is the relevance of the implicit bias of interpolators to the development?
>
> Thanks for the interesting question. We would like to refer to Section 3.4 for a discussion on different interpolators. In particular, this phenomenon is not specific to the max-$\ell_2$-classifier (motivated below), but the result can for example be extended to different interpolators such as the max-$\ell_1$-classifier.
>
> >Please motivate the choice of the robust max-margin solution as written in Eq. 4.
>
> The definition of the robust max-$\ell_2$-margin classifier given in equation 4, is the typical definition of the robust max-$\ell_2$-margin classifier. As explained in Section 3, we focus on the robust max-$\ell_2$-margin classifier as SGD /GD on the adversarial logistic loss converges to it (Liu et al. 2020) for separable data.
>
> >The statement of Theorem 3.1 needs to be rephrased. It was going in the direction of saying "the $epsilon_{te}$-robust error ..?" but ended up saying "the following holds"
>
> We made this choice to simplify sentence structure.
>
> >It appears you can safely move the definitions of $\Phi_{min}$ and $\Phi_{max}$  to the statement of the theorem.
>
> This choice was made to keep the theorem statement compact and more readable.
>
> >Equation 5 is not really usable as an equation with $\tilde{\Phi}$ unspecified. Part (a) as presented only serves the body of the paper to establish a monotonic relationship with the robust training budget $\epsilon_{tr}$, and as such, I recommend to present it as such.
>
> Equation 5 with $\tilde{\Phi}$ has two functions: first it shows that the robust error is monotonically increasing in $\epsilon_{tr}$ and given a dataset $D$, it specifies the exact relation between the robust error and $\epsilon_{tr}$. More specifically, we find by Equation 5 that the robust error is exponentially monotonically increasing in $\epsilon_{tr}$.  Hence, we believe that Equation 5 is valuable.
>
> >Please indicate in the statement of the theorem for part (b) that it assumes a fixed $\epsilon_{tr}=\epsilon_{te}$.
>
> There might have been an misunderstanding. We would like to emphasize that our theorem explicitly holds for $\epsilon_{tr} \neq \epsilon_{te}$ and hence it is not true that it assume $\epsilon_{tr}=\epsilon_{te}$. This is in fact the key feature of our result. As we elaborate in Section 3.2 and 6, t in contrast to general belief, for directed attacks in the low sample size regime, training with a *smaller* $\epsilon_{tr}$ than $\epsilon_{te}$ results in a classifier with a higher robust accuracy as depicted in Figure 3a. Hence, it is key that our results hold for $\epsilon_{tr} \neq \epsilon_{te}$.

---

> > ### Author Response · Authors · 2022-11-11
> > **Response to reviewer Section 3b**
> >
> > >Why the large values of  $\epsilon_{tr} \in [0,5]$ for the experiments in this section? It seems that for the experiments on images $\epsilon_{tr} \in [0,0.5]$
> >
> > The values of \epstrain and \epstest in the Figures are chosen relative to the separation $r$ and the variance parameter $\sigma$. Note that for any $\epsilon_{tr}$, $\epsilon_{te}$, $\sigma$, $d$ and $n$ that satisfy the assumptions in the main theorem, the bounds hold. However, we want to stress that we cannot compare the $\epsilon_{tr}$ and $\epsilon_{te}$ parameters from the linear case with the ones from the image experiments as we consider different attacks and we do not know $d$ and $r$ in the image experiments case.
> >
> > >the solution of adversarial training => $\epsilon_{te}$-robust margin ~ Eq. (1)
> >
> > We implemented the change of formulation.
> >
> > >$\epsilon_{tr}$ appears nowhere near Equation 7, yet  $Susc(\theta;\epsilon_{te})$  is defined as the $\epsilon_{tr}$-attack-susceptibility. I recommend you recall Equation 2 explaining how $\epsilon_{tr}$ is implicit in the training objective to obtain the $\theta$ in Equation 7.
> >
> > Thank you for finding this typo. It should indeed read “the $\epsilon_{te}$- susceptibility score $Susc(\theta;\epsilon_{te})$” instead of the $\epsilon_{tr}$-susceptibility score $Susc(\theta;\epsilon_{tr})$.
> >
> > >$θ^{\epsilon_{tr}}$  is used before it was defined.
> >
> > Could you point us to the line where it appears before it is defined in Equation 4? We were not able to find what you mention.
> >
> > **Nitpicking.**
> > Thanks, we implemented the changes

---

> > ### Comment · Reviewer_7vDF · 2022-11-12
> > **Follow up - 3 part A**
> >
> > Thank you for the detailed follow ups. I hope you're finding this to be a constructive exchange.
> >
> > I'm still concerned with how the notions of signals and class information are presented. Will revisit this point in a later follow up.
> >
> > The paragraph titled Robust-classifier could still use rewriting: the first sentence about implicit bias was unexpected given the flow of this section, and the discusssion immediately moves on. I asked about relevance to request a clarification within this paragraph. Writing "we focus on the robust max-ℓ2-margin classifier as SGD /GD on the adversarial logistic loss converges to it (Liu et al. 2020) for separable data" is much more informative than "we obtain Eq(4). This has been shown in (Liu)."
> >
> > Theorem 3.1 needs to be rephrased, because the second sentence "for any $\epsilon_{te} \geq 0$ .. (the) robust error .. the following holds" with the highlighted (the) is confusing. Perhaps you can move the $\epsilon_{te} \geq 0$ to the first sentence, then the second sentence can be "For the robust error on test samples from ... the following holds".
> >
> > Regarding Eq(5), is it possible to rewrite it as an inequality with an upper/lower bound, possibly with a concentration bound? Would it help to unpack $\Phi$ to better expose the exponential monotonicity?
> >
> > Thanks for clarifying the gap between $\epsilon_{te}$ and $\epsilon_{tr}$. Would it be better to also move the qualification to the subscript in $\widehat{\theta}^{\epsilon_{tr}}$?

---

> > > ### Author Response · Authors · 2022-11-16
> > > **Follow up 2 - 3 part A**
> > >
> > > >Thank you for the detailed follow ups. I hope you're finding this to be a constructive exchange.
> > >
> > > Thanks a lot for your timely response. We really appreciate the time you’re taking to understand our paper and point to formulations that may still be unclear.
> > >
> > > >I'm still concerned with how the notions of signals and class information are presented. Will revisit this point in a later follow up. The paragraph titled Robust-classifier could still use rewriting: the first sentence about implicit bias was unexpected given the flow of this section, and the discusssion immediately moves on. I asked about relevance to request a clarification within this paragraph. Writing "we focus on the robust max-ℓ2-margin classifier as SGD /GD on the >adversarial logistic loss converges to it (Liu et al. 2020) for separable data" is much more informative than "we obtain Eq(4). This has been shown in (Liu)."
> > >
> > > That is a good point, indeed the transition from the previous paragraph and the first sentences were not well formulated, thanks for catching that. We suggest the following rewriting, close to what you are proposing: “We study a classifier that is the solution of running gradient descent on the adversarial logistic loss. A long line of work (Soudry et al., 2018; Ji & Telgarsky, 2019; Chizat & Bach, 2020; Nacson et al., 2019; Liu et al., 2020) studies the implicit bias of (S)GD on the (standard) logistic loss and separable data. In particular, they show directional convergence to the max-margin solution. For the adversarial logistic loss and linear models in particular, (S)GD converges to the robust max-l2-margin solution ...”
> > >
> > > >Theorem 3.1 needs to be rephrased, because the second sentence "for any $\epsilon_{the}≥0$ .. (the) robust error .. the following holds" with the highlighted (the) is confusing. Perhaps you can move the $\epsilon_{the}≥0$ to the first sentence, then the second sentence can be "For the robust error on test samples from ... the following holds".
> > >
> > > You are right, the phrasing is suboptimal. It should instead read:
> > > “Assume d-1>n. For test samples samples from $P_r$ and any $0 \leq \epsilon_{te} < r/2$, the following holds for the $\epsilon_{te}$ robust test error of the classifier result from adversarial training with $\epsilon_{tr}$.”
> > >
> > > >Regarding Eq(5), is it possible to rewrite it as an inequality with an upper/lower bound, possibly with a concentration bound? Would it help to unpack Φ to better expose the exponential monotonicity?
> > >
> > > Eq 5 is a strict equality, there is no concentration needed - an upper and lower bound for $\Phi$ is given via sandwich inequalities for $\phi$ in Equation (6). Unpacking or even plotting $\Phi$ as a function of $\epsilon_{tr}$ is difficult, as it depends on many quantities such as $r, \phi_{max}$ etc. and one would need to make arbitrary choices and approximations that would muddle up the theorem. We thought the easiest way to parse the theorem is to consider concrete settings and plot the theoretical predictions (as we do in Fig 3 a), c)).
> > >
> > > >Thanks for clarifying the gap between $\epsilon_{te}$ and $\epsilon_{tr}$. Would it be better to also move the qualification to the subscript in $\theta^{\epsilon_{tr}}$?
> > >
> > > Do you mean we should use a subscript instead of a superscript? We don’t think it makes a big difference, and it would just result in a double subscript which is arguably a bit harder to read?

---

> > ### Comment · Reviewer_7vDF · 2022-11-13
> > **Signal direction and class information**
> >
> > As discussed earlier, those two notions are not formalized which creates lots of confusion. I attempt to provide some pointers to help improve this situation:
> > - Signal direction
> >   - How does this relate to the normal direction with respect to the data manifold? This notion has been used repeatedly in the context of adversarial robustness; see the references below.
> >   - By perturbing data in such direction, which the authors refer to as reducing class information, is it equivalent to make inputs more confusing to the model such that it could equally belong to more than one class?
> >   - If so, how does that relate to universal adversarial perturbations?
> > - Class information
> >   - How does this relate to salient features, which can be visualized using saliency maps?
> >   - Could you equivalently say that the proposed attacks specifically add noise to those parts of the input?
> >
> > References:
> > - Zhang, Wenjia, Yikai Zhang, Xiaoling Hu, Mayank Goswami, Chao Chen, and Dimitris N. Metaxas. "[A Manifold View of Adversarial Risk](https://proceedings.mlr.press/v151/zhang22h.html)." In International Conference on Artificial Intelligence and Statistics, pp. 11598-11614. PMLR, 2022.
> > - Lin, Wei-An, Chun Pong Lau, Alexander Levine, Rama Chellappa, and Soheil Feizi. "[Dual manifold adversarial robustness: Defense against lp and non-lp adversarial attacks](https://proceedings.neurips.cc/paper/2020/hash/23937b42f9273974570fb5a56a6652ee-Abstract.html)." Advances in Neural Information Processing Systems 33 (2020): 3487-3498.
> > - Yu, Bing, Jingfeng Wu, Jinwen Ma, and Zhanxing Zhu. "[Tangent-normal adversarial regularization for semi-supervised learning](https://openaccess.thecvf.com/content_CVPR_2019/html/Yu_Tangent-Normal_Adversarial_Regularization_for_Semi-Supervised_Learning_CVPR_2019_paper.html)." In Proceedings of the IEEE/CVF Conference on Computer Vision and Pattern Recognition, pp. 10676-10684. 2019.

---

> > > ### Author Response · Authors · 2022-11-16
> > > **Signal direction and class information Answer**
> > >
> > > Dear reviewer, thank you for the suggestions and pointers to the literature. We aim to clarify our definition of directed attacks in the following.
> > >
> > > First a general addendum, in hope to clarify the definition of directed attacks and as mentioned in the last sentence of the directed attack paragraph in Section 2 and the second sentence of the directed attacks paragraph in Section 3: a crucial property of directed attacks is that the direction (i.e. for additive perturbations, i.e. $x’ = x+ \epsilon \delta$, we mean the one-dimensional subspace where $\delta$ lies in) of the resulting perturbation is not calculated dependent on the “current model”. Instead, the set of perturbations is restricted to directions attacking the Bayes optimal classifier, or equivalently, the “ground truth”. We highlighted this fact in the updated manuscript in the directed attacks paragraph in Section 2.
> > >
> > > In case the reviewer is familiar with the formalism of Ilyas et al. 2019, we attempt another clarification using the concept of useful features: the *signal* direction in the distribution is the span of the collection of all the useful features (mathematically the ones that $E_{(x, y) \sim P}[ y \phi(x)] > \xi$). For example, in our linear setting, the only useful feature is $x_1$ and the directed attack sets in Equations 3 and 9 all reduce the strength of the useful feature, i.e. the separation in the first coordinate, from $r$ to $r-2 \epsilon_{te}$.
> > >
> > > > Signal direction: How does this relate to the normal direction with respect to the data manifold? This  notion has been used repeatedly in the context of adversarial robustness; see the >references below.
> > >
> > > First, note that as mentioned above, the signal is a concept that is crucially linked to the conditional distribution of $y|x$. On the other hand, the notion of a manifold in the referred papers are properties of the marginal distribution on $x$. Therefore, these notions are inherently different. In particular, directed attacks can be both on-manifold and off-manifold perturbations and do not necessarily have to be normal wrt. to the data manifold.
> > >
> > > >By perturbing data in such direction, which the authors refer to as reducing class information, is it equivalent to make inputs more confusing to the model such that it could equally belong to more than one class? This is the right intuition (see first paragraphs of this response). Consistent directed attacks result in perturbations that effectively reduce the class information sample but not to an extent that the Bayes optimal robust classifier wrongly classifies the perturbed sample. If so, how does that relate to universal adversarial perturbations?
> > >
> > > Universal attacks are attacks that fool many state-of-the art classifiers. This stands in contrast to directed attacks that are *independent* of any particular model but depend on the *data distribution*. For a particular dataset, a directed attack *may* still qualify as a universal attack despite not being specifically designed as such. On the other hand, universal attacks that can fool many state-of-the art classifiers may be independent of the ground truth classifier and are hence not directed attacks.
> > >
> > > *Class information*
> > > >How does this relate to salient features, which can be visualized using saliency maps? Could you equivalently say that the proposed attacks specifically add noise to those parts >of the input?
> > >
> > > Salient features and their maps are defined with respect to a specific model unlike directed attacks. Indeed, the salient features of the Bayes optimal classifier coincide with the pixels that a directed attack perturbs.  Finally, our directed attacks, however, do not add “noise” to the input but are adversarial. Hence, directed attacks effectively reduce the information about the class in the input or in other words reduce the strength of the useful features for classification.

---

> > > > ### Comment · Reviewer_7vDF · 2022-11-16
> > > > **Appreciate the clarification**
> > > >
> > > > Thanks for clarifying this crucial aspect of the work.
> > > >
> > > > As the authors would agree, I presume, the Bayes optimal classifiers as well as the work of Ilyas et al. (2019) were only mentioned in the related work section on Page 9. I can see in the revised manuscript that the authors now mention Bayes optimal classifiers upon describing directed attacks on Page 3. I am afraid I don't think this is adequate positioning for effective presentation to the wider ML community.
> > > >
> > > > Specifically, speaking of Bayes optimality seems essential, IMHO, to make sense of what is meant by "signal" or "information." In addition, I can see many related works ~~in the citation list of~~ citing Chen et al. (2020), some of which must be discussed with others warranting even a brief mention.
> > > >
> > > > This is not to say I don't see the value in the present submission, but it cements my conviction that the presentation, and perhaps some of the formalisms and experiments in turn, would need significant revision.
> > > >
> > > > I urge the authors to continue this important work, incorporating the Bayes formalism from the get go, clarifying how it relates to the robust max-margin classifier analyzed, and how it can be used to instantiate meaningful attacks in the context of image classification.
> > > >
> > > > Perhaps the proposed attack can be formalized as maximizing the information loss under a given perturbation budget, e.g., information-efficient / Bayes-optimal attacks (**Update**: I can see the authors already used this description upon presenting Eq.3). I strongly recommend you double-check for similar ideas in the adversarial robustness literature. Please also make sure to distinguish your analysis from similar analyses that are also grounded in the linear case, as in follow-up works to Chen et al. (2020).

---

> > > > > ### Author Response · Authors · 2022-11-17
> > > > > **Major revisions?**
> > > > >
> > > > > We appreciated the clarification questions by the reviewer so far and it seems that we have answered and resolved them by now. The last comments however are more generally about adding citations and some general comments about necessary revision that are less clear and concrete. We argue that we disagree with the reasoning behind the low score that a significant revision is necessary or helpful. However we added a few more  clarifications and are open to potentially adding a few more citations if the reviewers feel strongly about some and if they could point them out more specifically. We would appreciate some clarifications by the reviewer about their left-over concerns.
> > > > >
> > > > > >perhaps some of the formalisms and experiments in turn, would need significant revision.​​ I urge the authors to continue this important work, incorporating the Bayes formalism from the get go,
> > > > >
> > > > > The Bayes optimality viewpoint on directed attacks does not change anything in the actual formalism nor in the experiments, in our humble view. Just to make sure that there is no misunderstanding: By Bayes optimal classifier we mean the (unconstrained) population minimizer of the empirical 0-1 loss, equivalent to the *ground truth* for generative classification models. We had mentioned this explicitly in the introduction even in the first submitted version. To be clear, there is nothing really “Bayesian” going on here, hence no major new formalism has to be introduced.  In the next revised manuscript, we make clear from the beginning that the direction of directed attacks are independent from the current model.
> > > > >
> > > > > If you still have concrete “significant revisions” in mind that are needed in the “formalism or experiments”, we would very much appreciate it if you could be more specific. We will then upload another revised version on Friday 18/11.
> > > > >
> > > > > >clarifying how it relates to the robust max-margin classifier analyzed,
> > > > >
> > > > > The robust max-margin classifier is defined using perturbation sets that are in turn determined by the perturbation type - this is the exact place where our definition of directed attacks comes in (and the only time where the perspective of Bayes optimality comes in).
> > > > > In the linear setting that we analyze, the *ground truth* (in our generative model this is equal to the Bayes optimal classifier), which is a linear classifier. The robust max-margin classifier in general is the solution of adversarial training on the robust adversarial logistic loss. We added this additional clarification to Section 3.
> > > > >
> > > > > > and how it can be used to instantiate meaningful attacks in the context of image classification.
> > > > >
> > > > > This is in fact exactly what we did in the paper - the theory existed first, and we used the following recipe to successfully find meaningful attacks for object recognition tasks (for simplicity binary):
> > > > >
> > > > > a).   Identify the distinguishing factors for different objects  (such as texture, shape etc.)
> > > > >
> > > > > b).   come up with a natural corruption that would occlude / blur these distinguishing factors (examples are blur or occlusion attacks that we explored in the paper, but color attacks on objects could also work e.g. for fruits etc.)
> > > > >
> > > > > If you disagree with any of the above points and still strongly believe that a much bigger revision is needed to improve the paper above the acceptance threshold for you, it would be great if you can leave concrete comments about the concerns that are left.
> > > > >
> > > > > >In addition, I can see many related works in the citation list of Chen et al. (2020), some of which must be discussed with others warranting even a brief mention.
> > > > >
> > > > > We are not sure which papers in the bibliography of Chen et al. (2020) the reviewer is referring to. Obviously Chen contains a lot of citations that are generally related to adversarial training (of which we cite Szegedy, Goodfellow) - however it is impossible to cite every paper due to space constraints . The most related work to us, that would be crucial to cite for positioning our results, are, in our view, related to
> > > > >
> > > > > - accuracy vs. robustness tradeoff and mitigations such as self-training (from Chen’s list we already cite Tsipras, Raghunathan, Ilyas, Zhang 19/20, Stutz and more.)
> > > > > There are two more in Chen’s list that we do not cite Dohmatob et al. ‘19 (on the trade-off) and Najafi et al. ‘19 - however the other cited papers share the main messages from these papers. If the reviewers feel strongly about adding them, we can.
> > > > >
> > > > > - Since our attacks are not the standard lp-attacks, we find it important to cite other attacks than lp-norm bounded attacks that have been considered in the literature (from Chen’s list we already cite Desfooli + many more such as Engstrom, Eykholt, Kang, Logan, Laidlaw that are not in Chen)
> > > > >
> > > > > We cannot identify a certain subtopic or message you feel is not represented/mentioned in our related works.  If this is one of the main reasons to stick with your score, please let us know what’s missing for you content-wise.

---

> > > > > > ### Comment · Reviewer_7vDF · 2022-11-17
> > > > > > **Clarification of concerns**
> > > > > >
> > > > > > I edited my previous comment to clarify: I meant recent papers citing Chen et al.
> > > > > >
> > > > > > To assuage the author concerns, I am only evaluating the manuscript presented. As a representative of some fraction of the ML community, I have put considerable time trying to digest the work as presented, and have shared my take over 2-3 iterations. Luckily, though, my take wouldn't constitute a single point of failure for the verdict on this submission. I will make sure to engage in the reviewer discussion phase and revisit my position in light of what other reviewers have to say.
> > > > > >
> > > > > > Regarding **positioning** and citations, I mean the body of literature that enters the picture for me through our follow up conversation. Specifically, I'm referring to:
> > > > > >
> > > > > > >- Attias, Idan, Aryeh Kontorovich, and Yishay Mansour. "Improved generalization bounds for robust learning." In Algorithmic Learning Theory, pp. 162-183. PMLR, 2019.
> > > > > > >- Dobriban, Edgar, Hamed Hassani, David Hong, and Alexander Robey. "Provable tradeoffs in adversarially robust classification." arXiv preprint arXiv:2006.05161 (2020).
> > > > > > >- Min, Yifei, Lin Chen, and Amin Karbasi. "The curious case of adversarially robust models: More data can help, double descend, or hurt generalization." In Uncertainty in Artificial Intelligence, pp. 129-139. PMLR, 2021.
> > > > > > >- Dong, Chengyu, Liyuan Liu, and Jingbo Shang. "Data Quality Matters For Adversarial Training: An Empirical Study." arXiv preprint arXiv:2102.07437 (2021).
> > > > > > >- Xing, Yue, Qifan Song, and Guang Cheng. "On the generalization properties of adversarial training." In International Conference on Artificial Intelligence and Statistics, pp. 505-513. PMLR, 2021.
> > > > > > >- ~~Chen, Jinghui, Yuan Cao, and Quanquan Gu. "Benign Overfitting in Adversarially Robust Linear Classification." arXiv preprint arXiv:2112.15250 (2021).~~
> > > > > > >- Mendonça, Marcele OK, Javier Maroto, Pascal Frossard, and Paulo SR Diniz. "Adversarial training with informed data selection." In 2022 30th European Signal Processing Conference (EUSIPCO), pp. 608-612. IEEE, 2022.
> > > > > >
> > > > > > Regarding directed attacks in the image context:
> > > > > > > I looked again at Section 4 and I could not see how the applied motion blur and illumination corruptions can be regarded as instances of directed attacks -- except for the similarity in the obtained trade-off curves to those obtained in Section 3. (Let me also mention that it wasn't clear to me how the attack budget manifests in the choice of size for the blur kernel).
> > > > > > >
> > > > > > > I wonder if conventional attacks, e.g., norm-bounded perturbations or even adversarial patch attacks, would also exhibit similar trade-off curves if applied in this context. This would strengthen/weaken the qualification of motion blur and illumination as directed attacks.
> > > > > >
> > > > > > Nit:
> > > > > > - Appendix D.3: the verb form of applying a convolution filter is to convolve.

---

### Official Review · Reviewer_bvDH · 2022-10-25

**Confidence:** 4
**Correctness:** 3
**Technical Novelty And Significance:** 3
**Empirical Novelty And Significance:** 2
**Recommendation:** 6

**Clarity, Quality, Novelty And Reproducibility:**

The paper is well written and easy to follow. The ideas are presented well and in a clean manner. While the main results of this paper go against general ML knowledge, some of it can also be considered intuitive. The authors provide some details about the experiments they have done but more details are required to completely reproduce the results.

**Strength And Weaknesses:**

Strengths
1. The results presented in this paper are interesting and does indeed caution user from blindly applying standard ML techniques for adversarial training.
2. The theoretical aspect of the paper is sound and I prefer the way the authors draw inspiration from the linear classifiers and extend that to non-linear ones.
3. The experimental results are also neat and clearly convey the message the authors are trying to provide.

Weaknesses
1. While the results are neat, I would have also liked to seen a more detailed discussion regarding in which scenario these learnings would be useful. The experimental results are for image classifiers which usually require a large amount of data both for training from scratch and for transfer learning. Thus it is unclear to me where the gain is useful. As the authors themselves mention, one of the main reasons for this degradation would be catastrophic overlearning which is intuitive on its own.
2. One potential way to strengthen the findings of this paper would be to have a relation on when this robust error increase happens in terms of d and n. That aspect is currently missing from this paper.

**Summary Of The Paper:**

The authors tackle the problem of robust training. They show that when the training set is small robust training can hurt the robust performance of the model. They start by theoretically showing this result for linear classifiers. Using these results they also design directed attacks and show that robust error can indeed increase when the dataset is small.

**Summary Of The Review:**

This paper cautions ML practitioners against blindly applying ML techniques for robust learning. However, the scope of the application may not be large as most current ML techniques use a large dataset to train. This significantly takes away from the contributions of this paper.

---

> ### Author Response · Authors · 2022-11-10
> **Response to reviewer**
>
> Dear Reviewer,
>
> Thank you for finding our theoretical and empirical results interesting, sound and neat. In the following, we address the points that you mention as the main weaknesses of the paper and hope to clarify some of the questions. If you have any additional suggestions that would push the paper to acceptance for you we would be grateful to hear them during the discussion period. We now answer to the weaknesses you pointed out explicitly.
>
>
> >While the results are neat, I would have also liked to seen a more detailed discussion regarding in which scenario these learnings would   be useful. The experimental results are for image classifiers which usually require a large amount of data both for training from scratch and for transfer learning. Thus it is unclear to me where the gain is useful.
>
> Thanks for asking this specific question. We agree that many image classification tasks often require many samples even when using transfer learning. We would like to stress that low-sample regime refers to a low number of samples compared to the parameterization of the model (dimensionality d in the linear case). In particular, the sample size is still *large enough* both in our experiments and theoretical setting, to achieve good standard accuracy ($>90%$ in Figure 1) as noted in the caption of Figure 1 and the sentence before the contributions.. Hence, it is still a *relevant* low-sample regime that one may use in practice.
>
> >As the authors themselves mention, one of the main reasons for this degradation would be catastrophic overlearning which is intuitive on its own.
>
> We assume you are referring to “catastrophic overfitting” and our Discussion in Section 4.4.? What we wanted to stress in that paragraph is that in contrast to the catastrophic overfitting literature, which suggests that weaker attacks during training lead to worse generalization during test time, in our setting, weaker attacks (in terms of perturbation budget epsilon) actually lead to **better** generalization.
>
> >One potential way to strengthen the findings of this paper would be to have a relation on when this robust error increase happens in terms of $d$ and $n$. That aspect is currently missing from this paper.
>
> Indeed, the relation of the robust error increase crucially depends on the regime of $d$ and $n$ - in particular, the gap is larger in a regime where $d>>n$ but $n$ is still large enough to have good standard classification accuracy. This is formalized in our main theorem, and confirmed experimentally as depicted in Figure 1, Figure 3 b,c and Figure 5 c and 6c in the main text.
>
> Kind regards,
> the authors

---

> > ### Comment · Reviewer_bvDH · 2022-12-05
> > **Response to Authors**
> >
> > I thank the authors for their responses. After reading the responses and the other discussions in this thread I have decided to increase my score.

---

> ### Author Response · Authors · 2022-12-12
> **Answer on revised review**
>
> Dear reviewer,
> Thank you for the comments and discussion. We have gone through the mentioned references and will add a concise version of some of the following comments to the related work section of the manuscript to provide more context.
>
> 1. Uniform convergence generalization bounds for adversarial robustness in Attias et al. 2019 or asymptotic (in)consistency results Xing et al. 2021. However, these bounds are not fine-grained enough to predict the phenomenon that we establish in our paper.
>
> 2. Further discussion on the robust vs. standard accuracy tradeoff:  The message in Dobriban et al.  2020 is in spirit similar to Tsipras et al. (already cited in our paper), and discusses inherent tradeoffs between the optimal robust vs. standard classifier. Further, Min et al. 2021 show that adversarial training with large inconsistent $\ell_p$ adversarial attacks in noisy settings may increase the robust (test) error with the sample size $n$. In contrast to the above works, in this paper, we study a noiseless distribution with no inherent tradeoff (i.e. consistent perturbations) - that is there exists a classifier that is both optimal for the robust and standard (population) risk. For this setting, we compare robust generalization of different finite-sample *algorithms* - “adversarial training” with different perturbation budgets (including standard training for budget = 0).
>
> 3. Dong et al. 2021 and Mendonça et al. 2022 study the detrimental effect of low-quality/uncertain samples on the robustness-accuracy trade-off for adversarial training. Dong et. al. 2021 empirically shows that “low quality data'' may be the root cause behind the robustness-accuracy trade-off for $\ell_p$ attacks.  In contrast, we study adversarial training in a setting with “high quality data” and compare the robust error of adversarial with standard training.
>
> Further, we will revise the definition of directed attacks as requested and explain in more detail why the motion blur and illumination attacks are directed attacks.
>
> Kind regards,
> The authors

---

> > ### Comment · Reviewer_7vDF · 2022-12-13
> > **Positioning**
> >
> > Thanks for the follow up. I'm assuming this response was meant to me?
> >
> > While I'm not an expert on this particular thread, I hope the authors would take the time to reflect on how to best position their work in light of those closely related works. I do not wish for the authors to feel pressured to include extra citations or peripheral discussions mainly to obtain higher review scores.
> >
> > That said, it appears those references do approach the question of why adversarial training can hurt robust accuracy. I believe this discussion should be an integral part to the introduction and qualification of the scope of work and methodology adopted. I still think a more specific title is needed, especially if the authors are inclined to regard those works as essentially distinct.
> >
> > I understand there are differences between the perturbed samples used for adversarial training and low quality samples. In other words, whether the distribution of training data is inherently noisy (low quality), or was obtained by introducing noise to a supposedly noiseless distribution (high quality). However, there are also similarities. More importantly, it's not clear if those modeling assumptions apply equally well to any particular dataset.

---

### Official Review · Reviewer_SVdL · 2022-10-25

**Confidence:** 4
**Correctness:** 4
**Technical Novelty And Significance:** 3
**Empirical Novelty And Significance:** 4
**Recommendation:** 8

**Clarity, Quality, Novelty And Reproducibility:**

The paper is novel with high-quality theoretical analysis and empirical evaluation.

**Strength And Weaknesses:**

### Strength
- The paper presents very interesting observations that can potentially arouse wide attention for the practical usage of adversarial training.
- Solid theoretical analysis on linear classifiers with good intuitions provided.
- Experimental results focus on a customized Waterbirds dataset with motion blur and illumination and the results confirm the claims. CIFAR10 and hand-gesture dataset with square mask perturbations are also investigated.

### Weakness
- More types of direct attacks could be potentially evaluated, e.g., physical attacks on stop signs, to make the claims more convincing.

**Summary Of The Paper:**

This paper focuses on adversarial training with direct attacks that effectively reduce the information about the true classes. It presents interesting observations that adversarial training might hurt robustness performance under limited training sample sizes. Theoretical analysis with main theorems provided on linear classifier case study and the generalizability is discussed. Experimental results on the customized Waterbirds,  CIFAR10, and hand-gesture datasets support the claims.

**Summary Of The Review:**

Overall, the paper provides good observations on adversarial training with the directed attack. The authors provide solid theoretical analysis and sufficient empirical evaluations to support the claims.

---

> ### Author Response · Authors · 2022-11-10
> **Response to reviewer**
>
> Dear Reviewer,
>
> Thank you for appreciating our contributions. Regarding the following point
>
> >More types of direct attacks could be potentially evaluated, e.g., physical attacks on stop signs, to make the claims more convincing.
>
> Thanks for this nice suggestion. We would like to note that we do have mask attacks on CIFAR-10 (Appendix E) which are essentially inspired by sticker attacks/occlusions on traffic signs, one of the earlier papers on adversarial robustness beyond $\ell_p$ attacks. They reveal the same phenomenon as with the motion blur attacks. We moved them to the appendix for space reasons.
>
> Kind regards,
>
> the authors

---

### Decision · Program_Chairs · 2023-01-20

**Decision:**

Accept: poster

**Justification For Why Not Higher Score:**

The paper has certain weaknesses. The writing is also not as clear as it could be.

**Justification For Why Not Lower Score:**

Clearly over the acceptance threshold.

**Metareview: Summary, Strengths And Weaknesses:**

The paper shows that in the small-sample regime, adversarial training may actually hurt rather than help. This is demonstrated mathematically in the context of high-dimensional linear classification, and backed up by experiments on some image datasets with perceptible perturbations.

**Note From Pc:**

if the above contains the word "oral" or "spotlight" please see: "oral" presentation means -> notable-top-5% and "spotlight" means -> notable-top-25%. As stated in our emails, we are disassociating presentation type from AC recommendations